# Bootstrapping Hierarchical Autoregressive Formal Reasoner with Chain-of-Proxy-Autoformalization

**Qi Liu, Xinhao Zheng, Renqiu Xia, Qinxiang Cao, Junchi Yan**[*]
School of Computer Science & School of Artificial Intelligence, Shanghai Jiao Tong University
Shanghai Innovation Institute
{purewhite,void_zxh,xiarenqiu,caoqinxiang,yanjunchi}@sjtu.edu.cn
https://github.com/Purewhite2019/har_copa_main

## Abstract

Deductive formal problem-solving (D-FPS) enables process-verified, human-aligned problem-solving by implementing deductive solving processes within formal theorem proving (FTP) environments. However, current methods fail to address the misalignment between informal and formal reasoning granularity and suffer from inefficiency due to backtracking and error propagation. Moreover, the extreme scarcity of formal problem-solution pairs further hinders progress. For the first gap, we propose **HAR** (*Hierarchical Autoregressive Formal Reasoner*), a novel reasoning pipeline. HAR decouples informal-aligned drafting and detailed proving, and formulates solution construction as autoregressive generation with per-step feedback. Second, we propose **CoPA** (*Chain-of-Proxy-Autoformalization*), a data generation pipeline that cascades statement autoformalization, proof drafting, and proof search as a proxy autoformalization path. Experiments demonstrate significant improvements: trained on data bootstrapped by CoPA, HAR achieves superior performance on FormalMath500 ($15.50\% \mapsto 44.09\%$) and MiniF2F-Solving ($21.87\% \mapsto 56.58\%$) with lower computational budget. Explorations reveal promising directions in formal solution pruning and informal dataset denoising.

## 1 Introduction

Contemporary large reasoning models (LRMs) demonstrate superhuman performance in mathematical reasoning [1–3]. However, the lack of zero-false-positive process-level supervision limits their inference scalability [4] and training effectiveness [5, 6]. Deductive formal problem-solving (D-FPS) [7] addresses this by implementing problem-solving processes within formal theorem proving (FTP) environments, decoupling solving and verification, and enforcing deductive reasoning.

The transition from FTP to D-FPS introduces two core challenges. 1) **absence of an efficient and effective reasoning method**; 2) **extreme scarcity of formal problem-solution data**.

For the reasoning pipeline, we systematically analyze the design choices of existing FTP methods and identify two key gaps. First, the granularity between informal and formal reasoning is misaligned. For instance, while substituting $\sqrt{4}$ with 2 is trivial in informal reasoning, it requires a three-line formal proof. Conversely, computing the 100th Fibonacci number modulo 4 is easy in formal systems. Second, existing FTP paradigms are inefficient for deductive reasoning. Search-based approaches allocate enormous resources to backtracking. An agent can abandon the current path and return to a previous node to avoid dead ends. However, this is less essential for D-FPS. Whole-proof generation suffers from error propagation, where initial mistakes lead to cascading failures.

---

[*]Corresponding author. This work was in part supported by NSFC (92370201).

Therefore, we propose **HAR** (*Hierarchical Autoregressive Formal Reasoner*) for effective and efficient reasoning. HAR autoregressively generates solution steps to derive the answer without backtracking. Each reasoning step is decoupled into a high-level drafting step, which is aligned with informal reasoning, and a series of low-level proof searches to fill specific gaps. This approach combines the robustness for stepwise failure from search-based methods and the depth-priority from whole-generation methods. HAR shows significantly superior performance to existing methods, solving $44.09\%$ of FormalMath500 [8, 7] and $56.58\%$ of MiniF2F-Solving [9, 7] with the lowest budget. HAR even outperforms methods given ground-truth informal solutions.

For data bootstrapping, existing models lack task-specific knowledge and training. However, relevant FTP tasks such as statement autoformalization are well-explored. We propose **CoPA** (*Chain-of-Proxy-Autoformalization*), cascading statement autoformalization, proof drafting, and proof search as a proxy path to formalize problems and solutions. For quality filtering, we propose an explosion and vacuous check for problems, and a deductive check for solutions. Along with expert iteration, experiments demonstrate clear and consistent improvement in problem and solution autoformalization.

Formal problem and solution data produced by CoPA are process-level verified, which opens up interesting possibilities, e.g., **solution pruning** and **informal dataset denoising**. We design prototypes and conduct exploratory experiments to inspire future work. Solution pruning removes at most $23.41\%$ of tokens in formal solutions. Models trained on these data show on-par (and even superior) performance and shorter reasoning lengths during evaluation. Models fine-tuned on denoised informal data demonstrate clear improvement over those fine-tuned on the original data ($52.52\% \mapsto 59.19\%$).

## 2  Related Works

**Formal Reasoning.** Closely related works include those in formal theorem proving (FTP) and formal problem-solving (FPS). FTP methods aim to construct machine-verifiable proofs for formal mathematical statements. Roughly two main paradigms exist: *proof search* methods view the construction of a proof as a tree search process on potential proof paths [10]. Best-first search with value function as cumulated log-probability [11–17] or critic model [18, 19], and MCTS-based search [13] are used. Specifically, POETRY [15] proposes recursive best-first search for finer-grained search. Lean-STaR [16] proposes to augment formal step generation by informal thoughts and directly sample paths. *Whole-proof generation* methods [20–28] samples proofs conditioned on formal statements in one pass. DeepSeekProver-V1.5 [25] pioneers incorporating a truncate-and-resume mechanism to combine search and whole-generation. For D-FPS, prompt-based whole-solution generation [7] resembles whole-proof generation. The proposed HAR differs from existing methods by 1) splitting the generation of formal solutions into many smaller rejective sampling steps to avoid error propagation; 2) decoupling one formal step into an informal-aligned drafting and finer-grained gap filling; 3) removing backtracking to improve efficiency.

**Autoformalization.** Statement autoformalization aims at translating informal statements into formal ones while preserving semantics. Existing methods prompt general LLMs [29–31, 26], train autoformalizers with back-translation [32–34], and train with filtered autoformalization results [35, 26, 36]. Proof autoformalization [22, 37–40] focuses on generating formal proofs from informal ones using methods similar to statement autoformalization. However, they usually focus on proof correctness rather than semantic consistency. CoPA differs by 1) formalizing problems, answers, and solutions with carefully designed quality filtering; 2) cascading existing statement autoformalization, proof drafting, and proof search methods as a proxy path.

**Formal Data Generation.** Existing methods usually focus on FTP and can be roughly divided into three categories. Statement autoformalization [24, 35] enlarges the formal corpus by translating informal statements into formal ones and searching for their proofs. Expert iteration [12, 24] alternatively trains provers using cumulated proofs and uses trained models to generate more proofs. Statement synthesis [41, 42, 28] generates new statements by symbolic mutation [41] , forward reasoning [42], and conjecturer LLMs [28]. CoPA differs by 1) generating informal-formal aligned problems and solutions; 2) splitting solution drafting and proof gap filling.

## 3  Background

Beyond verifying theorem proofs, Lean 4 [43] can be used to provide process-level supervision in problem-solving processes under the framework of Formal Problem-Solving (FPS) and Deductive

DPS (D-FPS) [7]. Based on FPS, D-FPS decouples deducing the answer and validating its soundness into forward-solving and backward-proving processes, and enforces deductive reasoning in forward-solving. These modifications enhance the alignment between D-FPS solutions and informal solutions.

D-FPS define *problems* as predicates $P(\hat{A}) := (\forall_{(v:T)\in V}(v:T), \forall_{(h:\phi)\in\Phi}(h:\phi), (\psi \leftrightarrow A))[A \mapsto \hat{A}]$, and *answers* as propositions $\hat{A} :$ Prop, where $V, A, \Phi, \psi$ are the set of independent variables, the queriable, the set of hypotheses, and the conclusion, respectively. For example, a problem "If $mn = 4$ and $m + n = 5$, what is $|m - n|$?" with ground-truth "3", the above components can be $V = \{(a : \mathbb{R})(m : \mathbb{R})(n : \mathbb{R})\}, \Phi = \{(h_0 : mn = 4)(h_1 : m + n = 5)\}, \psi = (|m - n| = a)$. Therefore, it can be formalized into a predicate $P(\hat{A}) := \forall(a : \mathbb{R})(m : \mathbb{R})(n : \mathbb{R})(h_0 : mn = 4)(h_1 : m + n = 5), (|m - n| = a) \leftrightarrow \hat{A}$ with ground-truth answer $\bar{A} : a = 3$.

The process of problem-solving, i.e., deducing an answer $\hat{A}$ and proving $P(\hat{A})$, is formulated as a Markov Decision Process (MDP). Informally, a solution state maintains variables, hypotheses, deduced facts, and unproven goals, and a solution step (action) manipulates the state while preserving logical coherence. In parallel with formal theorem proving (FTP), solution states in D-FPS are concretized as proof states, and solution steps are tactic applications. For example, the forward state 0 ( Fig. 1 top) contains variables $(a : \mathbb{R}), (m : \mathbb{R}), (n : \mathbb{R})$ and hypotheses $(h_0 : mn = 4), (h_1 : m+n = 5), (h_{\text{answer}} : |m-n| = a)$. A solution step can deduce a new fact $h_{\text{square}} : (m-n)^2 = (m+n)^2 - 4mn$ based on the context and transit to the forward state 1 (Fig. 1 bottom right).

The forward state 0 is initialized from the problem. The agent iteratively executes solution steps $s_t$ to transform forward states until reaching a terminal state where an answer $\hat{A}$ and a proof that $V, \Phi, \psi$ can deduce $\hat{A}$ are submitted simultaneously (Fig. 1 left). Then, a backward-proving process is conducted to show $V, \Phi, \hat{A}$ can deduce $\psi$. The backward-proving part is a vanilla FTP task, and once the forward part finishes, the correctness can be checked by comparing the submission $\hat{A}$ with the ground-truth $\bar{A}$. Therefore, this paper focuses on the forward-solving part, i.e., deducing the final answer step-by-step.

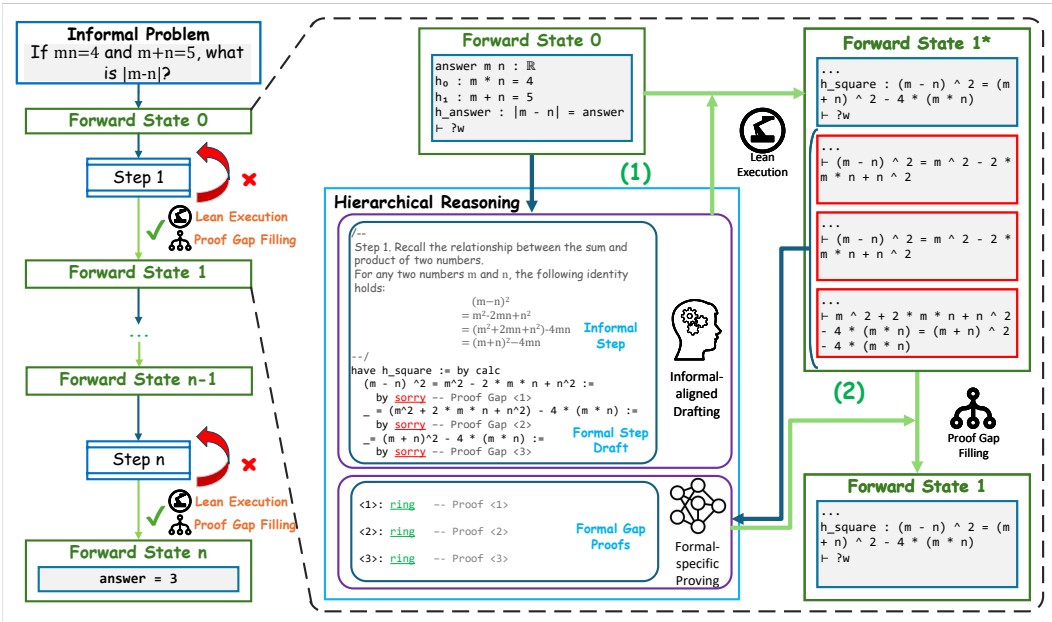

Figure 1: Pipeline of Hierarchical Autoregressive Formal Reasoner (HAR). HAR autoregressively transforms forward states without backtracking. Given a forward state, (1) HAR generates an informal step and a formal step draft. Lean 4 executes the draft, resulting in an intermediate state with unsolved goals. (2) HAR conducts proof searches to prove each goal, resulting in a new forward state. If the execution or proving fails, HAR retries until the step is successful or the step limit is exceeded.

# 4 Method

This section introduces the Hierarchical Autoregressive Formal Reasoner (HAR) for forward-solving in deductive formal problem-solving (D-FPS). Then, Chain-of-Proxy-Autoformalization (CoPA) is introduced for data generation. Finally, we present some intriguing explorations based on CoPA.

## 4.1 Hierarchical Autoregressive Formal Reasoner

### 4.1.1 Motivation

**Misaligned Granularity.** One significant discrepancy between informal and formal reasoning is their granularity. Some seemingly easy steps are nontrivial for formal reasoning, e.g., substituting $\sqrt{4}$ with 2, because real numbers are defined as equivalence classes of Cauchy sequences in formal mathematics. Conversely, computing the 100th Fibonacci number modulo 4 is easy. Therefore, directly generating formal steps from informal thoughts implicitly requires complex reasoning.

**Wasted Steps.** Proof search methods rely on tree search, where frequent backtracking causes a waste of reasoning steps. However, backtracking is necessary in FTP because it facilitates recovering from unprovable goals. These goals stem from unsafe tactics [44], e.g., aggressive application of `apply`, `exfalso`, and excessive `clear`. In D-FPS, many frequently used unsafe tactics are unavailable (discussed in Appendix A). Therefore, backtracking is not necessary. Whole-proof search methods suffer from error propagation. It generates all subsequent steps in one pass. Thus, the first erroneous step causes cascading failures of the whole proof, wasting all subsequent steps.

### 4.1.2 Design

The overall pipeline of the proposed Hierarchical Autoregressive Formal Reasoner (HAR) is in Fig. 1. The core idea is to treat the construction of a formal solution as a backtracking-free autoregressive search process. Each formal step is decomposed into an informal thought, a formal draft, and several fine-grained formal proofs.

**Hierarchical Reasoning.** To align the granularity between informal and formal reasoning, we split one formal step into a drafting step and several proof searches. The drafting step consists of an informal reasoning step and a formal step draft, which is generated following the informal step and may contain logical gaps. Then, proof searches fill the underlying gaps. Concretely, given an informal problem $P_i$ and the current forward state $S^{(t)}$, the next solution step is generated as follows.

- **High-level Drafting.** Sample an informal step $s_i^{(t)}$ from an LLM parameterized by $\theta$, i.e., $s_i^{(t)} \sim p_\theta(\cdot|P_i, S^{(t)})$. Then, sample a formal step draft $s_f^{(t)} \sim p_\theta(\cdot|P_i, S^{(t)}, s_i^{(t)})$. The draft is executed by Lean, resulting in the next state $S^{(t+1)}$, and unsolved goals $[\hat{S}_j^{(t+1)}]_{j=n}^m$ (corresponding to `sorry` placeholders in $s_f^{(t)}$), i.e., $(S^{(t+1)}, [\hat{S}_j^{(t+1)}]_{j=1}^n) = \text{Lean}(S^{(t)}|s_f^{(t)})$,

- **Low-level Proving.** For each unsolved goal $\hat{S}_j^{(t+1)}$, proof search is conducted to search for its proof $\hat{s}_{f,j}$. If all gaps are proven, the solution step $(s_i^{(t)}, s_f^{(t)}, [\hat{s}_{f,j}]_{j=1}^m)$ is returned and the current state is updated to $S^{(t+1)}$.

**Autoregressive Search.** The construction of the formal solution is formulated as an autoregressive search without backtracking. Concretely, an agent starts with some budget (max number of expansion attempts) $N_S$ and the initial forward state $S_0$. At state $S^{(t)}$, an expansion attempt includes generateing an informal solution step $s_i^{(t)}$, a formal step draft $s_f^{(t)}$, and the corresponding gap proofs $[\hat{s}_{f,j}]_{j=1}^m$. If the formal step draft is successfully executed by Lean and all proof searches succeed, the expansion succeeds and updates the solution state to $S^{(t+1)}$. Otherwise, the expansion fails. The agent will repeatedly retry until the budget is exhausted.

The agent submits answers by `exact` (implicitly submitting an answer proposition and its proof simultaneously) and terminates. If the submission is correct, the search succeeds; otherwise, the search restarts from $S_0$ with the remaining budget. If the budget is exhausted, the search fails.

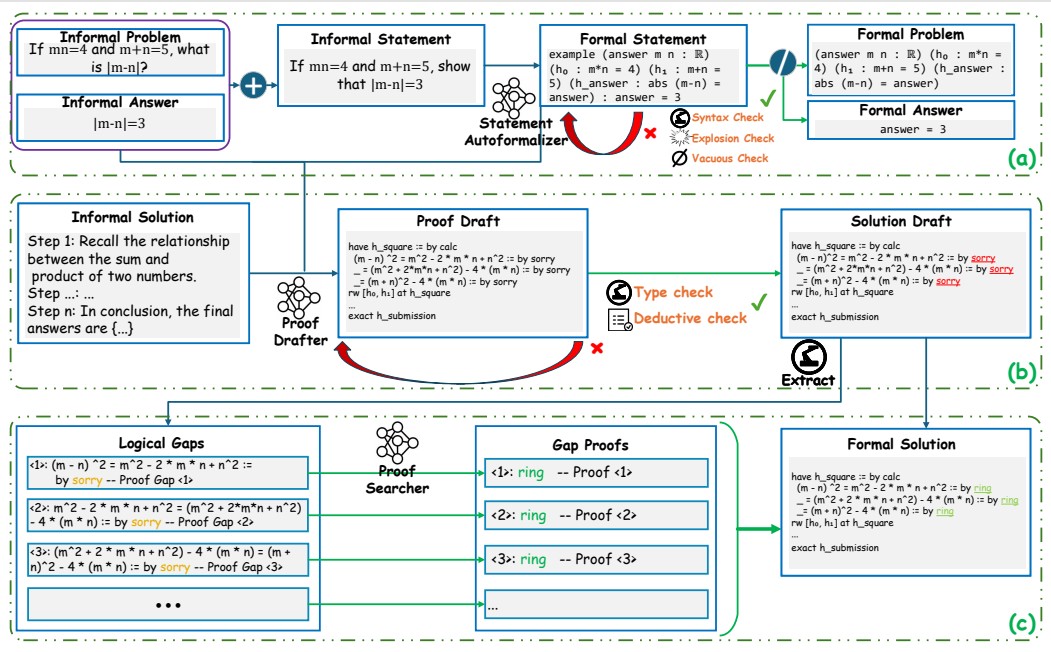

Figure 2: Demonstration of Chain-of-Proxy-Autoformalization (CoPA). (a) Given an informal problem and its informal answer, CoPA employs statement autoformalization as a proxy to formalize both the problem and its corresponding answer. Syntax check, explosion check, and vacuous check filter out low-quality formalizations; (b) Given an informal problem, its answer, solution, and formal statement, CoPA utilizes proof drafting as a proxy to draft a formal solution. Typecheck and deductive check filter out low-quality formalizations; (c) Given a solution draft, CoPA utilizes proof search to fill logical gaps, resulting in a completed formal solution.

## 4.2 Chain-of-Proxy-Autoformalization

The overall pipeline of Chain-of-Proxy-Autoformalization (CoPA) is in Fig. 2. The problem, answer, and solution autoformalization are reformulated as a chain of statement autoformalization, proof drafting [38], and proof search. Explosion, vacuous, and deductive checks are designed to improve data quality and ensure compatibility with D-FPS. To mine data towards the last nugget of gold, we design expert iterations specifically targeted at problem-solving. We also provide insights on potential explorations, including stepwise-verified solution pruning and informal solution denoising.

### 4.2.1 Problem Autoformalization

**Task Formulation.** *Problem autoformalization* is to translate an informal find-all problem $P_i$ and its ground-truth answer $\bar{A}_i$ into a formal problem $P_f$ with ground-truth answer $\bar{A}_f$. The translation should pass the syntax check and maintain semantics with $(P_i, A_i)$.

**Problem+Answer=Statement.** Statement autoformalization is used as a proxy for problem and answer autoformalization. Given an informal problem $P_i$ with answer proposition $\bar{A}_i$, concatenating $P_i$ and $\bar{A}_i$ can construct a statement that "if the conditions and conclusions are satisfied, then the ground-truth answer holds". Therefore, we use statement autoformalizers to get a formal statement $\forall_{j=1}^{m}(\chi_j : T_j), \bar{A}$, where $T_j$ can live in the `Type` and `Prop` universe. Based on the dependency of $\bar{A}$, we split out variables that $A$ depends on into a telescope $V = [v_j : T_j]_{j=1}^{n}$, hypotheses that $A$ depends into a telescope $\Phi = [h_j : \phi_j]_{j=1}^{p}$, and compose the remaining to $\psi$. Hence, we get a formal problem $P_f(\hat{A}) = (\forall_{j=1}^{n}(v_j : T_j), \forall_{j=1}^{p}(h_j : \phi_j), (\psi \leftrightarrow A))[A \mapsto \hat{A}]$ with ground-truth $\bar{A}$.

**Quality Filtering.** We filter out problems that fail in Lean syntax check, contain contradictory conditions, or are overly simple. Concretely, *explosion check* attempts to prove $\forall_{j=1}^{n}(v_j : T_j), \forall_{j=1}^{p}(h_j : \phi_j), \psi \rightarrow$ False and discard those proven. Relevant automation is detailed in Appendix E. If the proposition holds, the Modus Tollens rule concludes that there exist contradictions in conditions; thus,

arbitrary answer propositions hold by the principle of explosion. *Vacuous check* rejects if $A_f \in V \cup \Phi$ or $A_f$ is a clause in $\psi$ because the answer is too easy to derive, leaving the problem vacuous.

### 4.2.2 Hierarchical Solution Autoformalization

**Task Formulation.** *Solution autoformalization* is to translate an informal solution $s_i$ of problem $P_i, \bar{A}_i, P_f, \bar{A}_f$ into a formal solution $s_f$ which passes typecheck and is semantically consistent to $s_i$.

**Solution Draft $\approx$ Proof Draft.** Solution autoformalization is hierarchically divided into two stages: solution drafting and gap proving. Solution drafting aligns the granularity between formal and informal reasoning, and is implemented by the proxy of proof drafting. Forward solutions are a subset of proofs for $\forall_{j=1}^{n}(v_j : T_j), \forall_{j=1}^{p}(h_j : \phi_j), \psi \to \bar{A}$. Therefore, we use a proof drafter to draft $\mathring{s}_f$ for the statement. Then, *deductive check* filters out drafts if they contain tactics that change the target of proof goals, or do not simultaneously submit an answer and its proof with a direct `exact`.

**Solution Draft+Gap Proofs=Solution.** Given a solution draft $\mathring{s}_f$, the proof goals suppressed by `sorry` are extracted and a proof searcher constructs their proofs. If any proof search fails, the draft $\mathring{s}_f$ is rejected. Otherwise, the draft and the gap proofs are composed as the final formal solution $s_f$.

### 4.2.3 Expert Iteration

To unlock the full potential of existing data, we design an expert iteration pipeline that iteratively trains task-specific models and generates more data.

**Cold Start@Cycle** 0. Cold start is based on an augmentation[1] of MetaMathQA [45] and Numina-CoT [46]. We only use data from Numina-CoT and carefully filter out unsuitable problems, such as proving problems, synthetic ones, and those with corrupted answers or solutions. For finer-grained processing, we fine-tuned a Multilingual-E5-small [47] on the MATH [48] train set to label the subject and difficulty for each datapoint. We manually annotate formal statements and proof drafts for 100 demonstrations sampled from the MATH train set, then use DeepSeek-V3-12-26 [49] with subject-wise and level-wise 4-shot in-context learning to autoformalize statements and proof drafts. A proof searcher is trained with state-tactic pairs from Mathlib 4 [50] extracted by Pantograph [51] and the Lean-Workbook [35] dataset. The proof searcher is used to fill proof gaps. Resulting datapoints are categorized into 1) fails at statement autoformalization; 2) fails at proof drafting; 3) fails at proof gap filling; and 4) succeeds in all stages. Models are trained under the "next-stage" strategy: the statement autoformalizer is trained on datapoints from the third and fourth categories, the proof drafter is trained on only fourth-category data, and the proof searcher is trained on all succeeded gap proofs. Sophisticated methods are tested for filtering statement autoformalization data, but all perform worse than simple next-stage filtering, as detailed in Appendix D.

**Iteration@Cycle** $n$. In subsequent cycles, we use the Numina-1.5 [46] dataset and also filter out all proving problems, multiple-choice problems, and problems with corrupted answers. Model training follows the next-stage strategy on cumulated data from Cycle 0 to Cycle $n - 1$. At the end of each cycle, the datapoints of the fourth category are frozen and do not participate in the data generation of future cycles.

### 4.2.4 Explorations

Data produced by CoPA enables some intriguing explorations. Here we give a glimpse, and hopefully they can inspire readers of relevant directions.

**Solution Pruning.** Accompanied by inference-time scaling of large reasoning models (LRM), efficient reasoning [52–56] is drawing more attention. We demonstrate the feasibility of performing formally verified stepwise pruning on reasoning trajectories.

A formal solution $s_f = [s_{f,j}]_{j=1}^{m}$ may contain *redundant steps*, i.e., steps whose proposed hypotheses are not used in subsequent reasoning. We use Lean 4 Linter [43] to detect and remove redundant steps. Notably, this operation can be iteratively applied because there may be interdependencies among redundancies, where removing one subset may alter others.

**Dataset Denoising.** One of the most significant advantages of formal over informal reasoning is the process-verifiability, which can generate high-quality training data [57]. Here, we explore

---
[1]https://huggingface.co/datasets/yingyingzhang/metamath-qwen2-math

the possibility of repairing flaws in informal reasoning. Given an informal dataset of problems, answers, and solutions $\mathcal{D}_i = \{(P_i^{(j)}, \bar{A}_i^{(j)}), \boldsymbol{s}_i^{(j)}\}_{j=1}^m$, we run CoPA formalize them and get $\mathcal{D}_f = \{(P_f^{(j)}, \bar{A}_f^{(j)}), \boldsymbol{s}_f^{(j)}\}_{j=1}^m$. Then we translate formal solutions back, resulting in a denoised informal dataset $\bar{\mathcal{D}}_i = \{(P_i^{(j)}, \bar{A}_i^{(j)}), \bar{\boldsymbol{s}}_i^{(j)}\}_{j=1}^m$ with more trustworthy and consistent solutions.

## 5 Experiment

In this section, we answer the following research questions through comprehensive experiments.

- **Scalability of CoPA.** How scalable is CoPA with expert iteration on the datasets?
- **Effectiveness of HAR.** How effective is HAR over baselines and ablations?
- **Convergence of Best-first Search.** Will best-first search converge to whole-generation and autoregression as the expansion width approaches 1?
- **Solution Pruning.** Can solution pruning make training and inference more efficient?
- **Dataset Denoising.** Can formal-denoised data yields better fine-tuned models?

In each cycle of CoPA, 100 formalized statements are sampled and manually evaluated. HAR and its baselines are evaluated on FormalMath500 [7] and MiniF2F-Solving [7]. FormalMath500 is a formalized subset of MATH500 [48], consisting of 387 formal problems. MiniF2F-Solving is a refactored subset of MiniF2F [9], consisting of 375 formal problems[2]. See Appendix E, F,G,L, I for environment settings, training recipes, evaluation hyperparameters, prompt templates, and details about dataset denoising, respectively.

### 5.1 Baselines Methods

The following baselines for HAR are evaluated. All methods are fine-tuned from Qwen2.5-Math-7B [58] on data produced by CoPA.

**ICL** and **Hybrid CoT** are baselines for D-FPS introduced in [7]. ICL refers to in-context learning DeepSeek-V3 [49] with 10-shot demonstrations of informal problems, initial forward states, and ground-truth forward solutions. Hybrid CoT refers to additionally augmenting the forward solutions with aligned informal reasoning steps.

**Best-first Search (BFS)** resembles proof search in formal theorem proving (FTP), including a broad spectrum of works [59, 14, 18]. Given an informal problem $P_i$, a solution state $S_t$, it samples solution steps $s_t \sim p_\theta(\cdot|P_i, S_t)$ and conducts best-first search with value function $v_t = \sum_{t' \leq t} \frac{\log p_\theta(s_{t'} \mid P_{t'}, S_{t'})}{|s_{t'}|}$, i.e., the sum of normalized log-likelihoods of steps leading to $S_t$.

**Whole-Solution Generation (WG)** resembles whole-proof generation in FTP, including DeepSeek-Prover-V1 [24], DeepSeek-Prover-V1.5 [25], Goedel Prover [26]. Given an informal problem $P_i$, the initial solution state $S_0$, it samples the whole solution in one shot $\boldsymbol{s} \sim p_\theta(\cdot|P_i, S_0)$.

**Autoregressive Reasoning (AR)** is an ablation of the proposed HAR. Given an informal problem $P_i$, a solution state $S_t$, it iteratively samples solution steps $s_t \sim p_\theta(\cdot|P_i, S_t)$ until the step is correctly executed in Lean or the budget is exhausted. The solution construction is modeled as an autoregressive generation with per-step rejective sampling.

**Hierarchical Best-first Search (H-BFS)** and **Hierarchical Whole-Generation (H-WG)** are hierarchical variants of BFS [15] and WG. Instead of directly generating a complete step/solution, the model generates a step/solution draft (possibly with `sorry` placeholders), then conducts proof search to fill them.

**Hierarchical Solution Autoformalization (H-SA)** is introduced in Sec. 4.2.2. Given a formal statement (composed of the formal problem and its ground-truth answer) and a ground-truth informal solution, a proof drafter first translates the informal solution into a formal draft, and a proof searcher fills proof gaps. H-SA sets a strong baseline because it can be viewed as an augmented H-WG with ground-truth formal answers and informal solutions.

---

[2]We have conducted explosion check and vacuous check on these benchmarks. 9 of FormalMath500 and 5 of MiniF2F-Solving are filtered out in the following evaluation.

Table 1: Results of CoPA with expert iteration from Cycle 0 to Cycle 2. *Total* refers to the total number of datapoints to run CoPA; *Stages* represents the three stages of CoPA, where *Problem*, *Draft*, and *Gap* denote the success rate of problem autoformalization, solution drafting, and proof gap filling, respectively; *Problem Autoformalization* refers to the portion of autoformalized problems that are semantically correct from the human perspective; *FormalMath500* and *MiniF2F-Solving* denote the solving rate of applying proof drafting and proof gap filling on FormalMath500 and MiniF2F-Solving, respectively. Success rates and problem autoformalization accuracies are not directly comparable across cycles due to changes of data distribution. **Bold** highlights the best values.

| Cycle | Total | Stages | | | Problem Autoformalization | Formal-Math500 | MiniF2F-Solving |
|---|---|---|---|---|---|---|---|
| | | **Problem** | **Draft** | **Gap** | | | |
| 0 | 401,499 | 62.89% | 21.53% | 7.18% | 47% | 35.98% | 45.14% |
| 1 | 550,428 | 78.26% | 67.11% | **20.42%** | **62%** | 37.83% | 48.11% |
| 2 | 437,652 | **85.33%** | **76.96%** | 11.87% | 53% | **40.21%** | **50.27%** |

Table 2: Evaluation results of HAR against baselines and ablations. *Solved* represents the solving rates; *Budget* represents the total number of solution steps generated in one experiment; **Bold** highlights the best values among all experiments; Underlined values emphasize the best values in each ablative or comparative group. HAR demonstrates significant and consistent advantages in both solving rates and budgets, even compared with H-SA, which is enhanced with informal ground-truths.

| Data | Method | FormalMath500 | | MiniF2F-Solving | |
|---|---|---|---|---|---|
| | | Solved↑ | Budget↓ | Solved↑ | Budget↓ |
| None | ICL | 13.70% | 68382 | 21.87% | 65148 |
| | Hybrid CoT | 15.50% | 79153 | 21.60% | 81214 |
| Cycle 1 | BFS | 9.52% ± 0.57% | 28139 ± 104 | 9.64% ± 1.66% | 27712 ± 339 |
| | WG | 18.78% ± 0.22% | 18456 ± 92 | 24.95% ± 1.09% | 18853 ± 454 |
| | WG ($K_W = 16$) | 21.78% ± 0.12% | 35391 ± 153 | 28.83% ± 0.77% | 35895 ± 487 |
| | AR | 34.39% ± 0.78% | 20860 ± 233 | 44.41% ± 0.34% | 17814 ± 140 |
| | H-BFS | 13.32% ± 1.47% | 27777 ± 360 | 12.25% ± 1.47% | 27479 ± 203 |
| | H-WG | 36.77% ± 0.86% | 17039 ± 461 | 47.30% ± 1.01% | 16303 ± 388 |
| | H-WG ($K_W = 16$) | 40.04% ± 0.50% | 32134 ± 226 | 51.08% ± 0.44% | 30357 ± 212 |
| | H-SA | 37.83% | ∼ | 48.11% | ∼ |
| | **HAR** | 43.65% ± 1.35% | 18432 ± 344 | 55.50% ± 0.46% | 15069 ± 143 |
| Cycle 2 | H-SA | 40.21% | ∼ | 50.27% | ∼ |
| | **HAR** | **44.09%** ± 0.54% | 18273 ± 116 | **56.58%** ± 0.92% | 14754 ± 269 |

## 5.2 Results and Discussions

**Scalability of CoPA.** Results of expert iteration and evaluation are summarized in Table 1. We run expert iteration for three cycles. Cycle 0 is a cold start, while Cycle 1 and Cycle 2 are normal expert iterations. Expert iteration shows significant improvement in problem autoformalization. From Cycle 0 to Cycle 1, the accuracy of problem autoformalization improves from 47% to 62%. Notably, for Cycle 2, CoPA excludes successfully gap-filled data in Cycle 1 (*Gap*), which makes a direct comparison between Cycle 1 and Cycle 2 unfair. The portion of semantically correct problem autoformalizations on Cycle 1 results excluding *Gap* data is 49.30%. Therefore, the actual improvement from Cycle 1 to Cycle 2 is 49.30% ↦ 53%, which is also significant. Proof drafting and proof gap filling are evaluated on FormalMath500 and MiniF2F-Solving, where improvements are also significant. Due to the limited computation budget, we cannot afford to continue expert iteration and leave further scaling up as future work.

**Effectiveness of HAR.** Results are summarized in Table 2. Means and standard deviations of three runs are reported for ablations. We also evaluate WG and H-WG with max generation attempts $K_W = 16$ for a more intuitive comparison. Ablative improvements of hierarchical decoupling (from BFS, WG, and AR to their hierarchical variants H-BFS, H-WG, and HAR) are huge, especially the doubled solving rates between WG and H-WG (18.78% ↦ 36.77%, 24.95% ↦ 47.30%). As categorized in [7], one main error type of such whole-generation methods is logically flawed solution steps. Hierarchical decoupling makes per-step reasoning more reliable.

Table 3: Results on H-BFS on different expansion sizes $B \in \{1, 2, 4, 8, 16\}$. *Solved* represents the solving rates; *Avg. Len.* denotes the average lengths of correct solutions; *Budget* denotes the number of all generated steps; **Bold** highlights the best values. As $B$ lowers, H-BFS approaches HAR by increasing solving rates and solution lengths. On $B = 1$, H-BFS outperforms H-WG while still underperforming HAR.

| Method | $B$ | FormalMath500 | | | MiniF2F-Solving | | |
|---|---|---|---|---|---|---|---|
| | | Solved↑ | Avg. Len.↓ | Budget↓ | Solved↑ | Avg. Len.↓ | Budget↓ |
| H-BFS | 16 | 7.67% | 4.28 | 28824 | 6.76% | 4.40 | 28570 |
| | 8 | 15.34% | 4.76 | 27278 | 13.78% | 4.94 | 27227 |
| | 4 | 29.89% | 5.62 | 24588 | 31.35% | 5.82 | 24251 |
| | 2 | 39.68% | 6.71 | 21274 | 49.19% | 6.77 | 19197 |
| | 1 | 38.89% | 7.24 | 20197 | 50.00% | 8.08 | 17053 |
| H-WG | 1 | 36.77% | 7.10 | 16474 | 45.95% | 7.33 | 15760 |
| **HAR** | 1 | **41.80%** | 7.77 | 18902 | **54.86%** | 9.03 | 15271 |

Table 4: Experiment results of solution pruning. *Recursion* represents the number of pruning runs; *Train Size* denotes the number of tokens in the training set; *Solved* represents the solving rates; *Avg. Len.* denotes the average lengths of correct solutions; **Bold** highlights the best values.

| Recursion | Train Size↓ | FormalMath500 | | MiniF2F-Solving | |
|---|---|---|---|---|---|
| | | Solved↑ | Avg. Len.↓ | Solved↑ | Avg. Len.↓ |
| 0 | $5.85 \times 10^7$ | 34.39% ± 0.78% | 7.66 ± 0.18 | 44.41% ± 0.34% | 8.12 ± 0.33 |
| 1 | $4.92 \times 10^7$ | 34.30% ± 0.54% | 7.00 ± 0.01 | 44.59% ± 0.66% | 7.34 ± 0.22 |
| 2 | $4.64 \times 10^7$ | 34.57% ± 0.87% | 7.07 ± 0.21 | 44.50% ± 0.13% | **7.19** ± 0.05 |
| 3 | $4.54 \times 10^7$ | **34.83%** ± 0.33% | 6.88 ± 0.30 | **45.50%** ± 1.04% | 7.63 ± 0.11 |
| $\infty$ | $\mathbf{4.48 \times 10^7}$ | 34.48% ± 0.76% | **6.68** ± 0.16 | 45.32% ± 0.13% | 7.49 ± 0.20 |

Comparisons among BFS, WG and AR demonstrate AR's superiority in both solving rates and budget consumption. AR requires on-par budget of WG (21K vs. 18K, 18K vs. 19K), while reaching nearly double its solving rates ($18.78\% \mapsto 34.39\%$, $24.95\% \mapsto 44.41\%$). Compared with WG ($K_W = 16$), AR also has nearly half the budget and $1.5\times$ solving rates. Similar conclusions can be derived by comparing H-BFS, H-WG and HAR. One major improvement of AR (HAR) over WG (H-WG) is that it splits the autoregressive generation of the formal solution into many smaller rejective sampling steps: autoregressively sample the next solution step, reject if the Lean 4 typecheck (or the gap proof search) fails, until a logically correct step is generated. This finer-grained rejective sampling avoids a huge waste in WG: if one intermediate step fails, all following steps are wasted.

HAR also shows consistent improvement over expert iteration: between Cycle 1 and Cycle 2, the solving rates improve, and the budget lowers. Notably, in all the cycles, HAR even clearly outperforms H-SA, which incorporates ground-truth informal solutions.

More results and analysis are in Appendix B. Solving rates across different budgets validates HAR's Pareto-optimality over all baselines. Scaling experiments HAR and H-BFS to step limit $K_S = 320$ demonstrate HAR's stronger scaling capabilities, but HAR is limited by the LLM's context length. Extended evaluations on PutnamBench-Solving [7, 60] and MathOdessy [61] and experiments on Phi-4-mini-instruct [62] support the generalizability of HAR. Case studies can be found in Appendix J.

**Convergence of BFS.** We evaluate H-BFS with expansion widths $B \in \{1, 2, 4, 8, 16\}$. Results are shown in Table 3. Under a fixed budget, as $B$ decreases, the correct solutions become longer, the performance of H-BFS approaches and finally surpasses that of H-WG. This result aligns with the intuition that higher $B$ leads to breadth-first search and lower $B$ leads to depth-first search. H-BFS with $B = 1$ resembles H-WG and HAR. Its advantages over H-WG are 1) H-BFS explicitly receives the current solution state as input, while H-WG has to implicitly model solution states during generation; 2) Upon generating an incorrect step, H-BFS immediately exhausts the expansion width $B = 1$ and starts a new search, while H-WG wastes budget in generating the following steps. The disadvantage of H-BFS over HAR is also the early stop behavior. H-BFS's immediate restart on expansion failure prevents it from constructing longer solutions, thus underperforming on more complex problems. This hypothesis is supported by results that the average solution lengths of H-BFS with $B = 1$ are significantly lower than those of HAR (7.24, 8.08 vs. 7.77, 9.03).

**Solution Pruning.** Evaluation results of AR trained on Cycle 1 data recursively pruned for $R \in \{0, 1, 2, 3, \infty\}$ times are in Table 4. $R = 0$ denotes no pruning. $R = \infty$ indicates iterative pruning

Table 5: Experiment results of dataset denoising. *English Open*, *English MC*, *Chinese Gaokao*, *Chinese Other*, and *English Competition*, *Overall* represent the average accuracy on English open datasets, English multiple-choice datasets, Chinese gaokao collections, Chinese other datasets, English competition datasets, and all datasets, respectively; *Original*, *Direct SFT*, *Denoised SFT* represents the original model, the model fine-tuned using the original informal solution, and the model fine-tuned using the denoised informal solution, respectively; **Bold** highlights best values; Cyan and Purple indicate increases and decreases over the original model.

| Model | English Open | English MC | Chinese Gaokao | Chinese Other | English Competition | Overall |
|---|---|---|---|---|---|---|
| Original | 52.56% | 76.63% | 29.04% | 54.05% | 25.40% | 48.36% |
| Direct SFT | 60.13%(+7.57%) | **77.27%**(+0.64%) | 27.90%(-1.14%) | 61.35%(+7.30%) | **26.25%**(+0.85%) | 52.52%(+4.16%) |
| Denoised SFT | **64.70%**(+12.14%) | 76.60%(-0.03%) | **45.54%**(+16.5%) | **70.20%**(+16.15%) | 25.85%(+0.45%) | **59.19%**(+10.83%) |

until no redundant steps can be removed. We report the means and standard deviations of three runs. Although pruned $23.42\%$ of solution tokens (between $R = 0$ and $R = \infty$), the solving rates do not show a significant difference ($34.39\%, 44.41\%$ vs. $34.48\%, 45.32\%$), and the average solution length is greatly lowered ($7.66 \mapsto 6.68, 8.12 \mapsto 7.49$). See Appendix H for more results and analysis.

**Dataset Denoising.** Results are in Table 5, where benchmarks are categorized according to Qwen2.5-Math [58]. More details and discussions are in Appendix I. Denoised SFT model shows significantly superior performance on most datasets with relative improvements between $\sim 4\%$ to $\sim 18\%$. However, it falls short on English multiple-choice benchmarks ($-0.67\%$) and English competition benchmarks ($-0.40\%$). This drop is intuitive because CoPA focuses on free-response problems rather than multiple-choice questions. We leave more fine-grained formulations, such as multiple-choice and true-false problems, for future work.

## 6 Conclusion

This paper presents a systematic approach to two core challenges current formal problem-solving faces: the absence of a systematic study of pipeline design and extreme data scarcity. For the first, we propose **HAR** (*Hierarchical Autoregressive Formal Reasoner*), an informal-aligned, efficient and effective method for D-FPS. HAR decouples high-level reasoning and low-level proofs, and autoregressively constructs solutions. For the second, we propose **CoPA** (*Chain-of-Proxy-Autoformalization*), a scalable data generation pipeline for D-FPS. CoPA bypasses existing models' lack of D-FPS priors by chaining well-explored FTP tasks, statement autoformalization, proof drafting, and proof search as a proxy path. Based on data from CoPA, we explore the possibility of **Solution Pruning** and **Dataset Denoising**. Comprehensive experiments validate the effectiveness of the proposed methods.

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

## A  More Discussions on D-FPS

Tactic-style theorem proving [63] in Lean 4 [43] can be viewed as a Markov Decision Process (MDP). A *proof state* $\{\Gamma_i \vdash U_i\}$ is maintained by the Lean 4 environment, where $\Gamma_i \vdash U_i$ are *proof goals*, $\Gamma_i$ is a *local context* consisting of a telescope of *declarations*, and $U$ is a *target* type. Actions in this MDP are *tactics*, which transform a proof state by substituting a proof goal with a finite set of new goals (the set can be empty). For example, the following is a proof state with a single goal $\Gamma_0 \vdash U_0$. The local context consists of declaration of variables (answer $m\ n : \mathbb{R}$) and hypotheses $(h_0 : m * n = 4), (h_1 : m + n = 5), (h_{\text{answer}} : |m - n| = \text{answer})$. The target type is a proposition answer $= 3$.

```
answer m n : ℝ
h₀ : m * n = 4
h₁ : m + n = 5
h_answer : |m - n| = answer
⊢ answer = 3
```

Let's apply the following `have` tactic.

```
have h_square := by calc
  (m - n) ^2 = m^2 - 2 * m * n + n^2 := by
    ring
  _ = (m^2 + 2 * m * n + n^2) - 4 * (m * n) := by
    ring
  _= (m + n)^2 - 4 * (m * n) := by
    ring
```

A new hypothesis $h_{\text{square}} : (m - n)^2 = (m + n)^2 - 4 * (m * n)$ is introduced into the local context.

```
answer m n : ℝ
h₀ : m * n = 4
h₁ : m + n = 5
h_answer : |m - n| = answer
h_square : (m - n) ^ 2 = (m + n) ^ 2 - 4 * (m * n)
⊢ answer = 3
```

The proof state is initialized with a *formal statement* and iteratively transformed by tactics. The formal statement corresponding to $\Gamma_0 \vdash U_0$ is

```
example
  (answer m n : ℝ)
  (h₀ : m * n = 4)
  (h₁ : m + n = 5)
  (h_answer : abs (m - n) = answer)
: answer = 3
:= by
  sorry
```

The formal statement is proven once transformed into an empty proof state $\{\}$.

Tactics should be *provability-reflecting* [44]: proving all generated new goals implies proving the original proof goal, i.e., the following sequent holds.

$$\frac{\begin{array}{c} \Gamma_1' \vdash U_1' \\ \Gamma_2' \vdash U_2' \\ \cdots\cdots \\ \Gamma_n' \vdash U_n' \end{array}}{\Gamma \vdash C} \text{PR}$$

Conversely, however, not all tactics are *safe* [44] or *provability-preserving*: proving the original goal implies proving generated sub-goals, i.e., the following sequent holds.

$$\frac{\Gamma \vdash C}{\begin{array}{c} \Gamma'_1 \vdash U'_1 \\ \Gamma'_2 \vdash U'_2 \\ \ldots\ldots \\ \Gamma'_n \vdash U'_n \end{array}} \text{PP}$$

For example, tactic `have` introduces and proves a new hypothesis using existing hypotheses. It is naturally provability-preserving, because the only difference between the original goal and the one sub-goal generated by `have` is the newly introduced hypothesis, which can be proven using existing hypotheses.

However, tactic `exfalso` changes the target to `False`. It is not provability-preserving (or *unsafe* [44]) because it destroys provability for goals whose hypotheses do not contradict.

The ground-truth $\bar{A}$ : `Prop` satisfies

$$\forall_{j=1}^n (v_j : T_j), \forall_{j=1}^p (h_j : \phi_j), (\psi \leftrightarrow \bar{A})$$

i.e., the following goal is provable

$$\{[(v_j : T_j)]_{j=1}^n :: [(h_j : \phi_j)]_{j=1}^p \vdash \psi \leftrightarrow \bar{A}\} \tag{1}$$

And the forward-solving state of D-FPS is initialized as

$$\{[(v_j : T_j)]_{j=1}^n :: [(h_j : \phi_j)]_{j=1}^p :: [(h' : \psi)] \vdash ?A, [(v_j : T_j)]_{j=1}^n :: [(h_j : \phi_j)]_{j=1}^p \vdash \text{Prop}\} \tag{2}$$

Therefore, by rewriting $\psi$ into $\bar{A}$, we obtain

$$\{[(v_j : T_j)]_{j=1}^n :: [(h_j : \bar{A})]_{j=1}^p :: [(h' : \psi)] \vdash ?A, [(v_j : T_j)]_{j=1}^n :: [(h_j : \phi_j)]_{j=1}^p \vdash \text{Prop}\} \tag{3}$$

Any unsafe tactic application that breaks provability will lead to the final answer $\hat{A}$ to be logically inequivalent with the ground-truth $\bar{A}$. Therefore, a primary type of unsafe tactic applications, which modifies the targets of proof goals, is prohibited in D-FPS. Subsequently, the necessity of backtracking in D-FPS is weaker.

# B More Experiment Results

## B.1 Computational Efficiency

We further analyze the computational efficiency of HAR, baselines, and ablations. We evaluate WG and H-WG with maximal generation attempts $K_W = 16$ and collect the solving rates and budgets for $K_W = 1, 2, \ldots, 16$. For BFS, AR, H-BFS, HAR, we set the step limit to $K_S = 80$ and collect the solving rates and budgets for $K_S = 1, 2, \ldots, 80$. The solving rate - budget curves are visualized in Fig. 3. For a detailed comparison, the interpolated data is summarized in Table 6. HAR demonstrates Pareto-optimality over all scenarios.

## B.2 Test-time Scaling

We set the max step limit $K_S = 320$ to evaluate the test-time scalability of Cycle 2 H-BFS and HAR on the FormalMath500 benchmark. The interpolated solving rates are in Table 7. HAR solves 47.88% of problems with 64727 search steps, while H-BFS solves 32.54% with 90533 steps.

H-BFS shows consistent scalability and plateaus after $\sim 70000$ steps. The solving rate of HAR rapidly increases and is saturated after 20000 steps. The saturated solving rate of HAR (47.88%) is significantly higher than that of H-BFS (32.54%).

We find that HAR plateaus due to a limited context length. All models are SFTed from Qwen2.5-Math-7B, which has a maximum context length of 4096 tokens. Complicated problems require longer CoTs, which necessitate a larger context length for the model to cover the solution state (encompassing more conclusions along the reasoning) and the next-step draft.

HAR experiment encounters 953 times of context length exceeding (experiment restarts upon an exception occurring), while H-BFS encounters 413 times. This corresponds to the previous observation that BFS with accumulated log-prob as value function tends to collapse to breadth-first search.

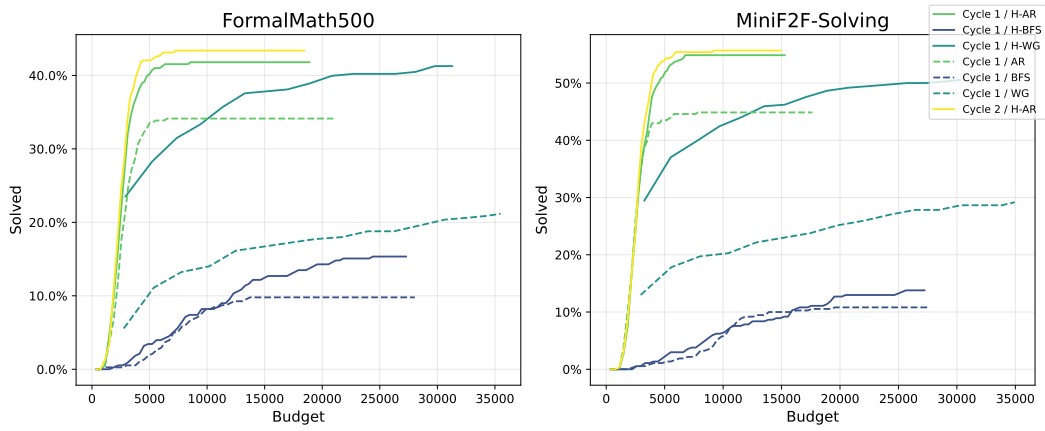

Figure 3: Solving rate - budget curves for HAR (Cycle 1, Cycle 2) and ablations.

Table 6: Solving rates of HAR over baselines and ablations across multiple budget settings. **Bold** highlights best values among all experiments; Underlined values emphasize the best values in each ablative or comparative group; "$\sim$" represents the corresponding budget setting is beyond the observation range.

| Benchmark | Cycle | Method | 1000 | 3000 | 5000 | 10000 | 15000 | 20000 | 30000 |
|---|---|---|---|---|---|---|---|---|---|
| **Formal-Math500** | 1 | BFS | 0.17% | 0.52% | 1.93% | 8.20% | 9.79% | 9.79% | $\sim$ |
| | | WG | $\sim$ | 6.06% | 10.32% | 13.96% | 16.72% | 17.79% | 20.13% |
| | | AR | 0.34% | 22.86% | 33.50% | 34.13% | 34.13% | 34.13% | $\sim$ |
| | | H-BFS | 0.00% | 0.78% | 3.44% | 8.20% | 12.46% | 14.29% | $\sim$ |
| | | H-WG | $\sim$ | 23.71% | 27.81% | 34.05% | 37.81% | 39.50% | 41.27% |
| | | HAR | 0.34% | 30.12% | 40.20% | 41.80% | 41.80% | $\sim$ | $\sim$ |
| | 2 | HAR | **0.85%** | **32.33%** | **42.06%** | **43.39%** | **43.39%** | $\sim$ | $\sim$ |
| **MiniF2F-Solving** | 1 | BFS | 0.00% | 0.54% | 1.23% | 5.80% | 10.00% | 10.81% | $\sim$ |
| | | WG | $\sim$ | 13.08% | 16.77% | 20.17% | 22.92% | 25.22% | 28.50% |
| | | AR | 0.00% | 35.29% | 43.51% | 44.86% | 44.86% | $\sim$ | $\sim$ |
| | | H-BFS | 0.00% | 0.60% | 2.21% | 6.40% | 9.01% | 12.70% | $\sim$ |
| | | H-WG | $\sim$ | $\sim$ | 35.30% | 42.71% | 46.18% | 48.97% | 50.49% |
| | | HAR | 0.00% | 35.67% | 51.76% | 54.86% | 54.86% | $\sim$ | $\sim$ |
| | 2 | HAR | **0.19%** | **38.81%** | **54.07%** | **55.68%** | $\sim$ | $\sim$ | $\sim$ |

Table 7: Solving rates of Cycle 2 HAR and H-BFS across multiple budgets (total number of solution steps generated in one experiment) settings. **Bold** highlights the best values among all experiments.

| Budget | FormalMath500 | | | | | | |
|---|---|---|---|---|---|---|---|
| | 2500 | 5000 | 10000 | 20000 | 40000 | 80000 | 90533 |
| H-BFS | 0.43% | 2.81% | 9.10% | 16.32% | 27.25% | 32.54% | 32.54% |
| HAR | **24.10%** | **44.39%** | **47.35%** | **47.88%** | **47.88%** | $\sim$ | $\sim$ |

## B.3  More Benchmarks

For a more comprehensive comparison, we evaluate HAR, baselines, and ablations on the PutnamBench-Solving [7, 60], which consists of 324 problems from undergraduate-level competitions. Moreover, we provide a pilot study on chaining the autoformalizer and problem-solver. We use the CoPA Autoformalizer to formalize 288 problems with non-empty labels in MathOdessy [61], resulting in 269 successful formalizations. HAR, baselines, and ablations are evaluated on this benchmark. Notably, in addition to settings in Appendix G, we also evaluate AR & HAR with step limit $K_S = 40$ and WG & H-WG with maximal generation attempts $K_W = 16$.

Table 8: Evaluation results of HAR, baselines, and ablations on PutnamBench-Solving and Math-Odessy. For PutnamBench-Solving, the numbers of solved problems are reported. Underlined values emphasizes the best values in each ablative or comparative group.

| Cycle | Experiment | PutnamBench-Solving | | MathOdessy | |
|---|---|---|---|---|---|
| | | Solved↑ | Budget↓ | Solved%↑ | Budget↓ |
| 1 | BFS | 1 | 25845 | 11.90% | 19490 |
| | WG | 0 | 13891 | 15.61% | 12773 |
| | WG ($K_W = 16$) | 0 | 27626 | 18.22% | 24865 |
| | AR ($K_S = 40$) | 2 | 12903 | 24.16% | 8613 |
| | AR | $\underline{2}$ | $\underline{25783}$ | $\underline{24.16\%}$ | $\underline{16773}$ |
| | H-BFS | 1 | 25842 | 15.24% | 19233 |
| | H-WG | 2 | 15909 | 26.39% | 12672 |
| | H-WG ($K_W = 16$) | 2 | 30981 | $\underline{28.62\%}$ | 24074 |
| | HAR ($K_S = 40$) | 2 | 12892 | $\underline{27.14\%}$ | 8417 |
| | HAR | $\underline{2}$ | $\underline{25772}$ | $\underline{28.25\%}$ | $\underline{16156}$ |
| 2 | HAR ($K_S = 40$) | 2 | 12893 | 27.88% | 8377 |
| | HAR | 2 | 25773 | 29.00% | 16038 |

Experiment results are shown in Table 8. Comparisons between AR & HAR ($K_S = 40$) and baselines validate AR&HAR's superiority over BFS & H-BFS and WG & H-WG across benchmarks. The only exception is observed on MathOdessy, where Cycle 1 H-WG ($K_W = 16$) slightly outperforms Cycle 1 HAR ($28.62\%$ vs. $28.25\%$) at the cost of $1.5\times$ budget.

However, it is worth noting that all methods almost fail on PutnamBench-Solving. We hypothesize that this is due to its out-of-distribution (OOD) nature. As stated in Appendix C, CoPA formalizes Numina-CoT and Numina-1.5 [46], which focuses on "Chinese high school math exercises to US and international mathematics olympiad competition problems". PutnamBench-Solving, on the other hand, focuses on undergrad-level competitions and is OOD with respect to the training set. The OOD curse in formal reasoning is more challenging since each formal step must be both semantically and syntactically correct. LLM usually fails to call functions or apply theorems without prior knowledge. We acknowledge that the current data domain is not broad enough and encourage future work to explore applying CoPA to formalize a wider range of datasets.

## B.4 Sensitivity to Base Models

To validate whether the impressive effectiveness is a flash in the pan on Qwen series models [64], we supervised fine-tune (SFT) HAR, baselines, and ablations based on Phi-4-mini-instruct [62] using Cycle 1 data, identical recipe and templates in Appendix F & L. The evaluation settings follow Appendix G. We also evaluate AR & HAR with step limit $K_S = 40$ and WG & H-WG with maximal generation attempts $K_W = 16$.

The results are in Table 9, which demonstrate similar patterns to results on Qwen2.5-Math-7B [65] and align well with our main claims. HAR consistently demonstrates Pareto-optimality across all baselines and ablations over all benchmarks.

## C  Limitations and Future Works

**Extension to Theorem Proving.** The proposed HAR model and CoPA data generator focus on the novel task of deriving answers for problems under the D-FPS framework. However, theorem proving and problem-solving are closely correlated. Moreover, given that HAR significantly outperforms strong ablative baselines in problem-solving tasks, its performance in theorem proving tasks is promising. Therefore, future work can extend HAR and CoPA to theorem proving data.

**More Expert Iteration.** Due to the limited computation resources, we only conduct CoPA expert iteration for three cycles (Cycle $0, 1, 2$). Given that statement autoformalization, solution drafting, proof gaps filling, and HAR all show increasing trends in Cycle 2, future work may continue cycling and seek the upper limit of expert iteration.

Table 9: Evaluation results of HAR, baselines, and ablations supervised fine-tuned (SFT) from Phi-4-mini-instruct [62] using Cycle 1 data. **Bold** highlights best values among all experiments; Underlined values emphasizes the best values in each ablative or comparative group.

| Experiment | FormalMath500 | | MiniF2F-Solving | | PutnamBench-Solving | | MathOdessy | |
|---|---|---|---|---|---|---|---|---|
| | Solved%↑ | Budget↓ | Solved%↑ | Budget↓ | Solved↑ | Budget↓ | Solved%↑ | Budget↓ |
| BFS | 7.41% | 28665 | 6.76% | 28230 | 1 | 25857 | 12.64% | 19408 |
| WG | 14.02% | 20295 | 21.08% | 20768 | 0 | 13987 | 13.38% | 14026 |
| WG ($K_W = 16$) | 15.61% | 40153 | 25.41% | 39366 | 0 | 27973 | 15.24% | 27356 |
| AR ($K_S = 40$) | 25.66% | 12198 | 31.35% | 11560 | 2 | 12893 | 24.54% | 8688 |
| AR | $\underline{26.72\%}$ | $\underline{23338}$ | $\underline{33.24\%}$ | $\underline{21464}$ | $\underline{2}$ | $\underline{25773}$ | $\underline{24.54\%}$ | $\underline{16808}$ |
| H-BFS | 11.90% | 27992 | 11.35% | 27591 | 1 | 25844 | 11.90% | 19679 |
| H-WG | 33.07% | 17457 | 41.35% | 17306 | 0 | 14306 | 25.28% | 12911 |
| H-WG ($K_W = 16$) | 38.10% | 33693 | 47.03% | 33742 | 1 | 30658 | 25.65% | 23894 |
| HAR ($K_S = 40$) | 41.80% | **10196** | 49.19% | **9247** | **3** | **12885** | 27.88% | **8328** |
| HAR | **42.06%** | $\underline{18956}$ | **49.73%** | $\underline{16687}$ | **3** | $\underline{25725}$ | **27.88%** | $\underline{16088}$ |

Table 10: Evaluation results of filtering methods. **Bold** highlights the best values among all experiments.

| Model | Method | Precision↑ | Recall↑ | Accuracy↑ | Kappa↑ |
|---|---|---|---|---|---|
| Qwen2.5-Math-7B-Instruct | $Prompt_1$+$Filter_1$ | 48.98% | 97.96% | 49.00% | 0.00 |
| | $Prompt_2$+$Filter_1$ | 52.17% | 97.96% | 55.00% | 0.11 |
| | $Prompt_1$+$Filter_2$ | 52.86% | 75.51% | 55.00% | 0.11 |
| | $Prompt_2$+$Filter_2$ | 51.06% | 97.96% | 53.00% | 0.08 |
| Qwen2.5-Math-72B-Instruct | $Prompt_2$+$Filter_1$ | 49.47% | 95.92% | 50.00% | 0.02 |
| | $Prompt_2$+$Filter_2$ | 49.47% | 95.92% | 50.00% | 0.02 |
| Rule-based | Problem Typechecked | 47.00% | **100.00%** | 47.00% | 0 |
| | Draft Typechecked | 70.59% | 51.06% | **67.00%** | **0.33** |
| | Proof Gap Filled | **72.73%** | 17.02% | 58.00% | 0.12 |

Table 11: Total variation distance (TVD) from the original dataset by filtering criteria.

| Method | TVD↓ | |
|---|---|---|
| | Difficulty | Subject |
| Problem Typechecked | 0.04 | 0.06 |
| Draft Typechecked | 0.12 | 0.12 |
| Proof Gap Filled | 0.31 | 0.16 |

**Data Domain.** CoPA expert iteration is conducted on the Numina-CoT [46] and Numina-1.5 [46] datasets, whose domain covers up to high-school competition-level mathematics. However, the expressiveness of Lean 4 covers mathematical research frontiers. Future work can explore data generation pipelines and reasoning methods that are more suitable for undergraduate-level, graduate-level, and even research-level mathematics and other subjects, along with recent progress in formalizations [66, 67].

# D    Filtering Problem Autoformalization

To filter out semantically inconsistent problems, we have designed and evaluated many model-based and rule-based methods.

**Methods.** For model-based filtering, we have evaluated two styles of prompts ($Prompt_{\{1,2\}}$) and two types of filtering metric ($Filter_{\{1,2\}}$) on Qwen2.5-Math-7B-Instruct [58] and Qwen2.5-Math-72B-Instruct [58]. For rule-based filtering, we have evaluated three types of filtering criteria: keep all successfully typechecked problems (*Problem Typechecked*), keep all problems whose solution drafts pass typecheck (*Draft Typechecked*), and only keep the problems whose proof gaps are all successfully filled (*Proof Gap Filled*).

**Accuracy.** We uniformly sample 100 datapoints from Cycle 0 data whose problems pass the Lean 4 typecheck and are manually labeled for semantic consistency. Evaluation results of the above filtering methods are shown in Table 10, where model-based methods all show extremely low accuracy ($\leq 55\%$) and Cohen's Kappa [68] ($\leq 0.11$). However, rule-based filtering shows significantly superior performance. *Draft Typechecked* reaches $67.00\%$ accuracy and $0.33$ Kappa, *Proof Gap Filled*'s precision is $72.73\%$.

**Distribution Shift.** We further compute the total variance distance (TVD) between distributions before and after filtering the Cycle 0 data, as shown in Table 11. From *Problem Typechecked*, *Draft Typechecked* to *Proof Gap Filled*, as the filtering becomes more strict, the distribution shift becomes more significant, especially regarding the difficulty.

Therefore, we adopt a filter to solve problems by the success of solution drafting (*Draft Typechecked*) as a trade-off between precision and distribution shift.

# E   Experiment Environment Details

In this project, the Lean 4 environment relies on the following open-source projects

- Lean 4 [43] `v4.15.0`
- Mathlib 4 [50] `v4.15.0`
- Aesop [44] `v4.15.0`
- Pantograph [51] `v0.2.25`
- Formal Problem-Solving [7] `39489d1f0c32b521845429e1cb26c48639d8f823`.

with the following options

- Lean 4 `maxHeartbeats: 0`
- Lean 4 `maxRecDepth: 100000`
- Lean 4 `tactic.hygienic: false`
- Lean 4 `pp.fullNames: true`
- Lean 4 `pp.funBinderTypes: true`
- Lean 4 `pp.piBinderTypes: true`
- Pantograph timeout: 300
- Pantograph imports: Mathlib, Aesop

For LLM fine-tuning, we use xTuner [69] `081c8ca874bdbf7a7f8cd0a9e4cba503eaaa7bba` with recipes detailed in Appendix F. For inference, we use vLLM [70] `0.6.0` with `bfloat16` type and prefix caching enabled. All experiments are conducted using a random seed $42$. For explosion check, we try each of `aesop`, `simp_all`, `abel`, `noncomm_ring`, `ring`, `module`, `nlinarith!`.

We sincerely appreciate the contributors of these awesome projects!

# F   Training Details

## F.1   Dataset Recipe

Given a dataset $\mathcal{D}_n$ produced by CoPA expert iteration at Cycle $n$, we construct training data for the following tasks:

- **Statement Autoformalization**. Datapoints whose formal solution drafts pass Lean 4 typecheck are used. Given an informal problem and its informal ground-truth answer, the model outputs a corresponding formal statement.
- **Solution Drafting**. Datapoints with successfully filled logical gaps are used. Input an informal problem, its informal ground-truth answer, its informal ground-truth solution, and a corresponding formal statement, and the model outputs a formal proof.

- **Next-Proof-Step Prediction**. Datapoints that have at least one logical gap filled are used. Moreover, proof state-tactic pairs from Mathlib 4 [50] extracted by Pantograph [51] and the Lean-Workbook [35] dataset are used. Input the current proof state; the model outputs the next proof step.
- **Next-Solution-Step Prediction**. Datapoints with successfully filled logical gaps are used. Input an informal problem and a current solution state; the model outputs the next solution step.
- **Next-Solution-Step Drafting**. Datapoints with successfully filled logical gaps are used. Given an informal problem and a current solution state, the model outputs the next formal solution step draft.
- **Whole-Solution Generation**. Datapoints with successfully filled logical gaps are used. Given an informal problem and its initial solution state, the model outputs a whole formal solution.
- **Whole-Solution Drafting**. Datapoints with successfully filled logical gaps are used. Input an informal problem and its initial solution state; the model outputs a formal solution draft.

Training data recipes of all methods are as follows:

- **CoPA**, **H-SA**: Statement Autoformalization, Solution Drafting, Next-Proof-Step Prediction;
- **BFS**, **AR**: Next-Solution-Step Prediction;
- **WG**: Whole-Solution Generation;
- **H-BFS, HAR**: Next-Proof-Step Prediction, Next-Solution-Step Drafting;
- **H-WG**: Next-Proof-Step Prediction, Whole-Solution Drafting.

### F.2 Training Hyperparameters

We use XTuner [69] for supervised fine-tuning (SFT) Qwen2.5-Math-7B [58] using the dataset recipes above (all tasks are uniformly mixed for training) and the following hyperparameters:

- Max Sequence Length: 8192
- Variable-length Attention: True
- Pack to Maximal Length: True
- Sequence Parallel Size: 1
- Batch size: 1
- Gradient Accumulation: 64
- Training Devices: 8
- Train Epochs: 3
- Optimizer: AdamW with learning rate $2 \times 10^{-5}$, $\boldsymbol{\beta} = (0.9, 0.999)$, weight decay 0, maximal gradient norm 1, warpup ratio 0.03 and `float16` mixed precision training.
- Learning Rate Scheduler: Warmup using LinearLR with start factor $10^{-5}$, then train using CosineAnnealingLR with $\eta_{\min} = 0.0$.

## G  Evaluation Details

### G.1  CoPA and HAR

Proof Gap Filling uses best-first proof search with a maximum number of step generation attempts $N_K = 16$, expansion width $B = 4$, temperature $T = 0.7$, and maximal sequence length $G = 256$ across experiments.

CoPA Expert Iteration. For each datapoint, statement autoformalization and solution drafting are tried at most 2 times until passing Lean 4 typecheck, each set temperature $T = 0.7$ and maximal sequence length $G = 4096$.

**ICL** and **Hybrid CoT** are not re-evaluated in this project. We utilize the evaluation results from [7] and further analyze their budget consumption.

**BFS** and **H-BFS** uses maximum number of step generation attempts $K_S = 80$, expansion width $B = 8$, temperature $T = 0.7$, and maximal sequence length $G = 2048$ by default. We also experiment with different expansion widths in Sec. 5.

**WG** and **H-WG** uses maximum number of whole-solution generation attempts $K_W = 8$, temperature $T = 0.7$, and maximal sequence length $G = 2048$.

**AR** and **HAR** uses maximum number of step generation attempts $K_S = 80$, temperature $T = 0.7$, and maximal sequence length $G = 2048$.

**Discussion.** We try our best to ensure fair comparisons. In the dataset constructed by CoPA, each formal solution contains 9.7 steps on average. Therefore, for search and autoregression methods, the maximum number of step generation attempts is set to 80; for whole-generation methods, the number of generation attempts is set to 8. Finer-grained stepwise budget consumption is presented in tables (*Budget*).

### G.2 Dataset Denoising

In the experiments of dataset denoising, all models are evaluated using the scripts of Qwen2.5-Math [58] in `https://github.com/QwenLM/Qwen2.5-Math` of commit `a45202bd16f1ec06f433442dc1152d0074773465`. We use the default evaluation script, where the random seed is set to 0, the temperature is set to $T = 0$, and the Top P is set to 1. The only difference is that we changed the system prompt to "You are a Lean 4 expert." when evaluating the SFT models to be consistent with the training process.

## H   More Results and Discussions on Solution Pruning

The effectiveness of solution pruning in Table 4 is mixed. For FormalMath500, there is no significant change in solving rates across pruning recursions, and average solution lengths decrease monotonically.

However, for MiniF2F-Solving, when $R = 0, 1, 2$, solving rates remain relatively stable and average solution lengths decrease monotonically; when $R = 3, \infty$, solving rates improve and the average solution lengths fluctuate. We hypothesize that this is because complex problems require longer chain-of-thoughts, which are limited by the Qwen2.5-Math-7B's context length (4096 tokens). Deeper pruning avoids redundant deductions that overflow the context length, thus facilitating deeper reasoning. Let $S_0, S_\infty$ denote problems solved in $R = 0, \infty$ experiments, respectively. The average solution length of $S_0 \backslash S_\infty$ is 9.82, while that of $S_\infty \backslash S_0$ is 11.85. The longest solution in $S_0 \backslash S_\infty$ has 17 steps, while $S_\infty \backslash S_0$ contains three solutions longer than 17 steps (19, 24, 29). This observation further supports improvements in both training and reasoning efficiency.

The distributions of solution lengths in training data are visualized in Fig. 4, where a clear distribution shift can be observed. Starting from an unimodal distribution (Recursion 0), the distribution of solution lengths gradually becomes bimodal, where double-step solutions become a new mode. We manually investigated the double-step solutions and found that they are almost all simple arithmetic problems and grade-school math. Lean has powerful tactics for these problems, such as `norm_num` and `linarith`. Solution of these easy problems collapses to the "guess-and-verify" pattern, i.e., directly guess an answer and verify it using automated tactics such as `linarith`. As visualized in Fig. 5 and manually checked by us, the collapsed problems are mainly grade school math (GSM) algebra problems.

These overly pruned solutions lack chain-of-thought reasoning, which may degrade reasoning capabilities [71]. However, in the main experiment, the model's performance does not show a statistically significant drop (and even improves), and the reasoning becomes much more efficient, reducing the number of reasoning steps by $\frac{1}{8}$. We hypothesize that this "free" efficiency improvement stems from the antagonism between removing CoT in GSM (which may reduce performance) and efficient reasoning for more difficult problems. We recommend that researchers apply solution pruning only to complex problems to avoid degrading CoT for simple ones. More detailed discussions and

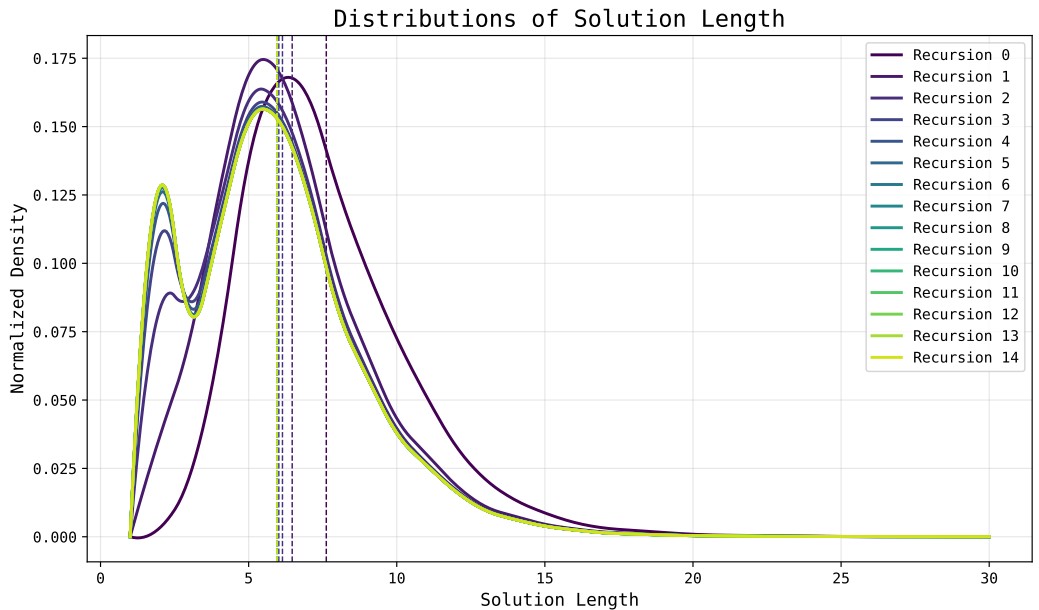

Figure 4: Distributions of solution lengths in training data.

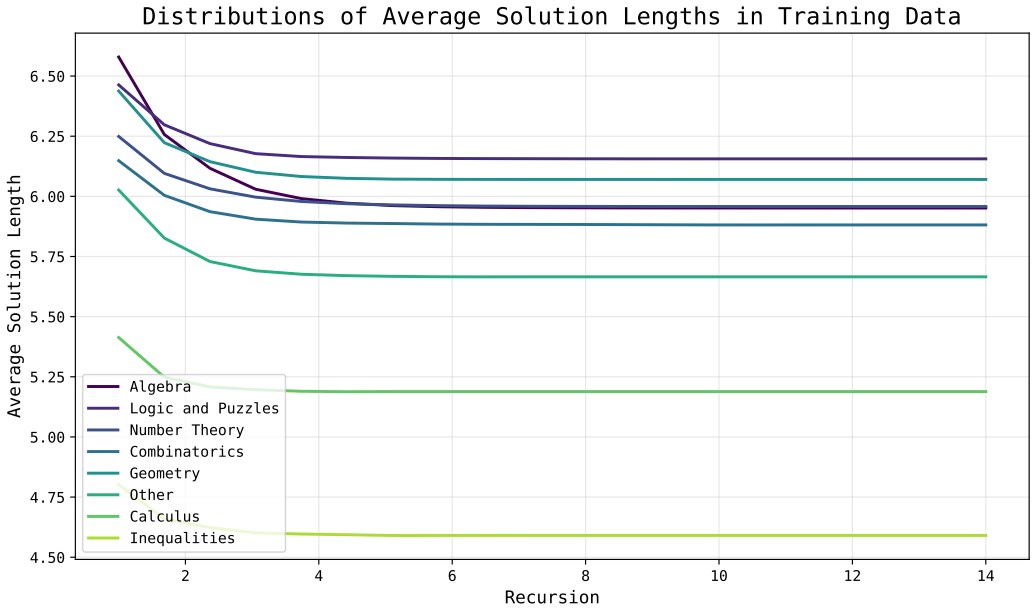

Figure 5: Distributions of average solution lengths by subject.

explorations are out of scope. We leave them for future work, and hope the above points inspire readers.

# I  Details and Discussions about Dataset Denoising

We randomly sample 8K datapoints from the Cycle 1 data. We then prompt DeepSeek-V3 to translate the solution drafts into informal solutions, given the informal problems, informal answers, and formal statements, with an explicit requirement to ensure semantic equivalence and no specific filtering for

Table 12: Detailed experiment results of dataset denoising. **Original**, **Direct SFT**, **Denoised SFT** represent the original model, the model fine-tuned using the original informal solution, and the model fine-tuned using the denoised informal solution, respectively; Accuracies are reported as a percentage; **Bold** highlights the best values among all experiments.

| Model | English Open Datasets | | | | | | |
|---|---|---|---|---|---|---|---|
| | GSM8K | MATH | SVAMP | ASDIV | MAWPS | CARP(EN) | TabMWP |
| **Qwen2.5-7B** | 66.9 | 62.6 | 70.7 | 74.1 | 69.2 | 50.3 | 57.8 |
| **Direct SFT** | 79.2 | 63.6 | **88.2** | 85.6 | 91.2 | **57.9** | 53.2 |
| **Denoised SFT** | **86.2** | **64.9** | 87.5 | **92** | **96.6** | **57.9** | **71.6** |

| Model | English Open Datasets | | | | English Multiple-Choice Datasets | | |
|---|---|---|---|---|---|---|---|
| | Minerva-MATH | Gaokao2023(EN) | OlympiadBench | CollegeMath | AQUA | SAT-Math | MMLU-STEM |
| **Qwen2.5-7B** | 16.5 | 46 | 32.7 | 31.4 | 67.7 | **90.6** | **71.6** |
| **Direct SFT** | 18.4 | 55.3 | **35.3** | 33.5 | 70.1 | **90.6** | 71.1 |
| **Denoised SFT** | **25** | **56.1** | 30.2 | **43.7** | **70.9** | 87.5 | 71.4 |

| Model | Chinese Gaokao Collections | | | | | Chinese Other Datasets | |
|---|---|---|---|---|---|---|---|
| | Gaokao2024(I) | Gaokao2024(II) | Gaokao2024(Mix) | Gaokao-Math-Cloze | Gaokao-Math-QA | CMath | CN-MiddleSchool |
| **Qwen2.5-7B** | 14.3 | 35.7 | 25.3 | 22 | 47.9 | 65.5 | 42.6 |
| **Direct SFT** | 14.3 | 28.6 | 26.4 | 16.9 | **53.3** | 70.2 | 52.5 |
| **Denoised SFT** | **35.7** | **50** | **38.5** | **50.8** | 52.7 | **77** | **63.4** |

| Model | English Competition Datasets | |
|---|---|---|
| | AIME24 | AMC23 |
| **Qwen2.5-7B** | **13.3** | 37.5 |
| **Direct SFT** | 10 | 42.5 |
| **Denoised SFT** | 6.7 | **45** |

fairness. See Appendix L.9 for the prompt template. Then, we fine-tune Qwen2.5-Math-7B [58] with the identical setting in Appendix F on the original informal solution and the denoised informal solution, respectively. The full evaluation results are in Table 12.

Although denoised SFT yields significant overall improvement, it falls behind on SVAMP, Olympiad-Bench, SAT-Math, MMLU-STEM, Gaokao-Math-QA, and AIME24. The causes might be:

- **Multiple-choice Questions (MCQs) and True-false Problems (TFPs).** CoPA focuses on free-response problems (FRP), and therefore rewrites TFPs/MCQs into FRPs in Numina-CoT, Cycle 0, and filters out MCQs in Numina-1.5, Cycle $n$. This distribution shift might leave these problems out-of-distribution, thus negatively impacting the SFTed models' performance on benchmarks of MCQs, including SAT-Math, MMLU-STEM, and Gaokao-Math-QA.

- **Simple math word problems (SMWPs).** SVAMP focuses on SMWPs about simple arithmetic taught in grades four and lower. Highly automated tactics like `norm_num` and `linarith` can directly solve them, thus their formalization might collapse to the "guess-and-verify" pattern, or even only a simple `norm_num`. Then, the informalization might contain a less detailed CoT compared to the original solution.

- **Out-of-distribution (OOD).** STEM problems, such as physics, biology, and astronomy, are hard to formalize and OOD to Numina-1.5. For problems in geometry, probability, and combinatorics, the main difficulty lies in understanding the abstract mathematical concepts and finding the expressions. Once formalized, most of them may collapse to simple arithmetic. This work focuses on problem-solving; therefore, theorem-proving ones are filtered out. Since MMLU-STEM consists of MCQs of STEM, OlympiadBench contains 25%+ physics problems and 21% about theorem-proving, which are relatively OOD to the SFT dataset and suffer from unstable performance.

- **Benchmark Size.** AIME24 only contains 30 problems, and the evaluation pipeline (Qwen2.5-Math's official repo) uses greedy decoding with temperature $T = 0$, sample number $n = 1$. This setting is prone to randomness in small benchmarks like AIME24; one sample can result in 3.3% performance variation.

## J  Case Studies

### J.1  Success Cases

**Informal Problem**

The graph of $f(x) = \frac{2x}{x^2 - 5x - 14}$ has vertical asymptotes $x = a$ and $x = b$, and horizontal asymptote $y = c$. Find $a + b + c$.

Answer: 5

**Initial Forward State**

```
answer a b c : ℝ
h₀ : ∀ (x : ℝ), x ≠ a → x ≠ b → x ^ 2 - 5 * x - 14 ≠ 0
h₁ : a ≠ b
h₂ : a ^ 2 - 5 * a - 14 = 0
h₃ : b ^ 2 - 5 * b - 14 = 0
h₄ : c = 0
h_answer : a + b + c = answer
⊢ ?w
```

**Submitted Answer**

```
answer = 5
```

**Formal Solution**

```
-- To find the vertical asymptotes of the function \( f(x) = \frac{2x}{x^2 - 5x -
   14} \), we need to determine where the denominator is equal to zero, as the
   function is undefined at these points.
have h_denominator_roots : a^2 - 5 * a - 14 = 0 ∧ b^2 - 5 * b - 14 = 0 := by --
   The roots of the denominator are \( a \) and \( b \)
  exact ⟨h₂, h₃⟩

-- 2. **Horizontal Asymptote:**
-- The horizontal asymptote of $f(x)=\frac{2x}{x^2-5x-14}$ is determined by the
   degrees of the polynomial in the numerator and denominator. Since the degree
   of the denominator is higher than the numerator, the horizontal asymptote is $
   y = 0$.
have h_horizontal_asymptote : c = 0 := by -- Horizontal asymptote is $y = 0$
  exact h₄

-- To find $a + b + c$, we need to find the sum of the roots $a$ and $b$ of the
   quadratic equation $x^2 - 5x - 14 = 0$.
-- According to Vieta's formulas, the sum of the roots of a quadratic equation $
   ax^2 + bx + c = 0$ is given by $-\frac{b}{a}$.
-- In our case, $a = 1$, $b = -5$, and $c = -14$. Therefore, the sum of the roots
   is:
-- \[ a + b = -\frac{-5}{1} = 5. \]
have h_sum_roots : a + b = 5 := by
  have h₅ : a ≠ b := h₁
  have h₆ : a ≠ b := h₅
  apply (mul_right_inj' (sub_ne_zero.2 h₅)).1
  nlinarith

-- 4. **Find the sum \(a + b + c\):**
--    - We have determined \(a + b = 5\) and \(c = 0\). Therefore, \(a + b + c = 5 +
   0 = 5\).
have h_sum : a + b + c = 5 := by
    linarith

have h_answer : answer = 5 := by
    linarith

exact h_answer -- submit

example
(answer : ℕ)
(h_answer : answer = Σ x ∈ (Σ n ∈ Nat.divisors 500, n).primeFactors, x)
: answer = 25 := by
```

```
-- First, we find the prime factorization of $500$:
-- $$500 = 2^2 \cdot 5^3$$
have h_prime_factorization : 500 = 2^2 * 5^3 := by
  norm_num [Nat.divisors]

-- The sum of the divisors of $500$ is given by the formula:
-- \[ A = (1 + 2 + 2^2)(1 + 5 + 5^2 + 5^3) \]
have h_sum_of_divisors : Σ n in Nat.divisors 500, n = (1 + 2 + 4) * (1 + 5 + 25 +
    125) := by
  norm_cast at h_answer h_prime_factorization ⊢

have h_sum_calculation : (1 + 2 + 4) * (1 + 5 + 25 + 125) = 7 * 156 := by
  norm_num [h_answer, h_prime_factorization, h_sum_of_divisors]

-- $A = 1092$
have h_A : Σ n in Nat.divisors 500, n = 1092 := by
  norm_num [h_answer, h_prime_factorization, h_sum_of_divisors, h_sum_calculation]

-- Now, factorize $A$:
-- \[1092 = 2^2 \cdot 3 \cdot 7 \cdot 13\]
have h_factorization : 1092 = 2^2 * 3 * 7 * 13 := by
  norm_num [h_answer, h_prime_factorization, h_sum_of_divisors, h_sum_calculation,
    h_A]

-- The distinct prime factors of $1092$ are $2$, $3$, $7$, and $13$.
have h_prime_factors : Nat.primeFactors 1092 = {2, 3, 7, 13} := by
  simp only [Nat.primeFactors, h_factorization, h_answer, h_sum_of_divisors,
    h_sum_calculation, h_A]
  norm_cast at h_factorization ⊢
  norm_cast at h_factorization ⊢
  norm_cast at h_factorization ⊢
  norm_num [Nat.primeFactorsList]

-- The sum of the distinct prime divisors is:
-- $$2 + 3 + 7 + 13 = 25$$
have h_sum_prime_factors : Σ x in Nat.primeFactors 1092, x = 2 + 3 + 7 + 13 := by
  simp [h_answer, h_prime_factorization, h_sum_of_divisors, h_sum_calculation, h_A,
    h_factorization, h_prime_factors]

-- Thus, the sum of the distinct prime divisors of $A$ is $\boxed{25}$.
have h_answer : answer = 25 := by
  norm_num [h_answer, h_prime_factorization, h_sum_of_divisors, h_sum_calculation,
    h_A, h_factorization, h_prime_factors, h_sum_prime_factors]

exact h_answer -- submit
```

### Informal Problem

Find the positive solution to $\frac{1}{x^2-10x-29} + \frac{1}{x^2-10x-45} - \frac{2}{x^2-10x-69} = 0$

Answer: 13

### Initial Forward State

```
answer x : ℝ
h₀ : 0 < x
h₁ : x ^ 2 - 10 * x - 29 ≠ 0
h₂ : x ^ 2 - 10 * x - 45 ≠ 0
h₃ : x ^ 2 - 10 * x - 69 ≠ 0
h₄ : 1 / (x ^ 2 - 10 * x - 29) + 1 / (x ^ 2 - 10 * x - 45) - 2 / (x ^ 2 - 10 * x -
    69) = 0
h_answer : answer = x
⊢ ?w
```

### Submitted Answer

```
answer = 13
```

**Formal Solution**

```
-- First, introduce a substitution to simplify the equation. Let:
-- \[
-- y = x^2 - 10x
-- \]
-- Then the given equation becomes:
-- \[
-- \frac{1}{y - 29} + \frac{1}{y - 45} - \frac{2}{y - 69} = 0
-- \]
have h_substitution : 1 / (x^2 - 10 * x - 29) + 1 / (x^2 - 10 * x - 45) - 2 / (x^2
    - 10 * x - 69) = 0 := h₄

-- Now, let's find a common denominator for the fractions on the left-hand side.
    The common denominator is \((a-29)(a-45)(a-69)\). Rewrite each fraction with
    this common denominator:
-- \[
-- \frac{(a-45)(a-69) + (a-29)(a-69) - 2(a-29)(a-45)}{(a-29)(a-45)(a-69)} = 0
-- \]
have h_common_denominator : (1 / (x^2 - 10 * x - 29)) + (1 / (x^2 - 10 * x - 45)) -
    (2 / (x^2 - 10 * x - 69)) = 0 := h_substitution

-- Since the denominators are not zero, we can combine the fractions:
have h_combined_fractions : (x^2 - 10*x - 45) * (x^2 - 10*x - 69) + (x^2 - 10*x -
    29) * (x^2 - 10*x - 69) - 2 * (x^2 - 10*x - 29) * (x^2 - 10*x - 45) = 0 := by
  field_simp [h₀, h₁, h₂, h₃, h₄] at h_common_denominator ⊢
  linarith

-- Now, simplify the numerator:
-- \[
-- (x^2 - 10x - 45)(x^2 - 10x - 69) + (x^2 - 10x - 29)(x^2 - 10x - 69) - 2(x^2 -
    10x - 29)(x^2 - 10x - 45)
-- \]
have h_numerator_expansion : (x^2 - 10*x - 45) * (x^2 - 10*x - 69) + (x^2 - 10*x -
    29) * (x^2 - 10*x - 69) - 2 * (x^2 - 10*x - 29) * (x^2 - 10*x - 45) = 0 := by
  linarith

-- Notice that each term in the numerator is a quadratic expression in the form of
    \( (y-a)(y-b) \). We can expand each term:
-- \[
-- (y-45)(y-69) = y^2 - 114y + 3015
-- \]
-- \[
-- (y-29)(y-69) = y^2 - 98y + 2001
-- \]
-- \[
-- 2(y-29)(y-45) = 2(y^2 - 74y + 1305) = 2y^2 - 148y + 2610
-- \]
have h_expanded_numerator : (x^2 - 10*x - 45)*(x^2 - 10*x - 69) + (x^2 - 10*x -
    29)*(x^2 - 10*x - 69) - 2*(x^2 - 10*x - 29)*(x^2 - 10*x - 45) =
    0 := by
  linarith

-- Expand each term in the numerator
have h_expanded_numerator_terms : (x^2 - 10 * x - 45) * (x^2 - 10 * x - 69) + (x^2
    - 10 * x - 29) * (x^2 - 10 * x - 69) - 2 * (x^2 - 10 * x - 29) * (x^2 - 10 * x
    - 45) = 0 := by
  linarith

-- Therefore, the equation simplifies to:
-- \[
-- \frac{0}{(y+29)(y+45)(y+69)} = 0
```

```
-- \]
have h_simplified_equation : (0) / ((x^2 - 10 * x - 29) * (x^2 - 10 * x - 45) *
    (x^2 - 10 * x - 69)) = 0 := by
  field_simp [h₀, h₁, h₂, h₃, h₄, h_answer, h_substitution, h_common_denominator,
    h_combined_fractions, h_numerator_expansion, h_expanded_numerator,
    h_expanded_numerator_terms]

-- Since the numerator is zero, the equation holds true for any $y$ that does not
    make the denominators zero.
have h_numerator_zero : 0 = 0 := by
  linarith

-- This confirms that the numerator is indeed zero, and the equation holds.
have h_denominator_nonzero : (x^2 - 10 * x - 29) * (x^2 - 10 * x - 45) * (x^2 - 10 *
    x - 69) ≠ 0 := by
  simp [h₀, h₁, h₂, h₃]

-- Conclusion: The equation is satisfied for all \( x \) such that the denominators
    are not zero.
-- Given that \( x \) must be positive, we need to check the values of \( x \) that
    make the denominators zero and exclude them.
have h_denominator_roots : ∀ x : ℝ, x^2 - 10*x - 29 = 0 ∨ x^2 - 10*x - 45 = 0 ∨
    x^2 - 10*x - 69 = 0 → x ≠ answer := by
  intro x h₅ h₆
  have h₇ : x ^ 2 - 10 * x - 29 = 0 ∨ x ^ 2 - 10 * x - 45 = 0 ∨ x ^ 2 - 10 * x -
    69 = 0 := h₅
  simp_all [h₀, h₁, h₂, h₃, h₄, h_answer, h_substitution, h_common_denominator,
    h_combined_fractions,
    h_numerator_expansion, h_expanded_numerator, h_expanded_numerator_terms,
    h_simplified_equation,
    h_numerator_zero, h_denominator_nonzero]

-- Since the numerator is zero and the denominator is non-zero, the equation is
    satisfied for all \(x\) except where the denominator is zero.
have h_solution : 0 = 0 := by
  linarith

-- Therefore, the positive solution to the original equation is:
-- \[
-- \boxed{13}
-- \]

have h_answer : answer = 13 := by -- The positive solution is 13
  nlinarith [h₀, h₁, h₂, h₃, h₄, h_answer, h_substitution, h_common_denominator,
    h_combined_fractions,
    h_numerator_expansion, h_expanded_numerator, h_expanded_numerator_terms,
    h_simplified_equation,
    h_numerator_zero, h_denominator_nonzero, h_denominator_roots, h_solution]

exact h_answer -- submit
```

**Informal Problem**

Assume that $x_1, x_2, \ldots, x_7$ are real numbers such that

$$x_1 + 4x_2 + 9x_3 + 16x_4 + 25x_5 + 36x_6 + 49x_7 = 1,$$
$$4x_1 + 9x_2 + 16x_3 + 25x_4 + 36x_5 + 49x_6 + 64x_7 = 12,$$
$$9x_1 + 16x_2 + 25x_3 + 36x_4 + 49x_5 + 64x_6 + 81x_7 = 123.$$

Find the value of $16x_1 + 25x_2 + 36x_3 + 49x_4 + 64x_5 + 81x_6 + 100x_7$.

Answer: 334

**Initial Forward State**

```
answer a b c d e f g : ℝ
h₀ : a + 4 * b + 9 * c + 16 * d + 25 * e + 36 * f + 49 * g = 1
h₁ : 4 * a + 9 * b + 16 * c + 25 * d + 36 * e + 49 * f + 64 * g = 12
h₂ : 9 * a + 16 * b + 25 * c + 36 * d + 49 * e + 64 * f + 81 * g = 123
h_answer : answer = 16 * a + 25 * b + 36 * c + 49 * d + 64 * e + 81 * f + 100 * g
⊢ ?w
```

**Submitted Answer**

```
answer = 334
```

**Formal Solution**

```
-- Define the equations
have h₀ : a + 4 * b + 9 * c + 16 * d + 25 * e + 36 * f + 49 * g = 1 := by -- $x_1 +
    4x_2 + 9x_3 + 16x_4 + 25x_5 + 36x_6 + 49x_7 = 1$
  linarith [h₀, h₁, h₂, h_answer]

have h₁ : 4 * a + 9 * b + 16 * c + 25 * d + 36 * e + 49 * f + 64 * g = 12 := by --
    Second equation
  linarith

have h₂ : 9 * a + 16 * b + 25 * c + 36 * d + 49 * e + 64 * f + 81 * g = 123 := by
  linarith

-- We subtract equation (1) from equation (2) and equation (2) from equation (3) to
    eliminate $x_1$ from the differences.
have h₃ : (4 * a + 9 * b + 16 * c + 25 * d + 36 * e + 49 * f + 64 * g) - (a + 4 * b
    + 9 * c + 16 * d + 25 * e + 36 * f + 49 * g) = 12 - 1 := by -- Subtract
    equation (1) from equation (2)
  linarith

-- Simplify to get $3x_1 + 5x_2 + 7x_3 + 9x_4 + 11x_5 + 13x_6 + 15x_7 = 11$
have h₄ : 3 * a + 5 * b + 7 * c + 9 * d + 11 * e + 13 * f + 15 * g = 11 := by
  linarith

-- Next, subtract the second equation from the third
have h₅ : (9 * a + 16 * b + 25 * c + 36 * d + 49 * e + 64 * f + 81 * g) - (4 * a +
    9 * b + 16 * c + 25 * d + 36 * e + 49 * f + 64 * g) = 123 - 12 := by
  linarith

have h₆ : 5 * a + 7 * b + 9 * c + 11 * d + 13 * e + 15 * f + 17 * g = 111 := by
  linarith

-- Calculate the next difference: $(5a + 7b + 9c + 11d + 13e + 15f + 17g) - (3a +
    5b + 7c + 9d + 11e + 13f + 15g) = 111 - 11$.
have h₇ : (5 * a + 7 * b + 9 * c + 11 * d + 13 * e + 15 * f + 17 * g) - (3 * a + 5 *
    b + 7 * c + 9 * d + 11 * e + 13 * f + 15 * g) = 111 - 11 := by
  linarith

have h₈ : 2 * a + 2 * b + 2 * c + 2 * d + 2 * e + 2 * f + 2 * g = 100 := by
  linarith

have h₉ : a + b + c + d + e + f + g = 50 := by
  linarith

-- Now, we find $y_4$ by adding this equation to $3y_2$:
have h₁₀ : 3 * a + 5 * b + 7 * c + 9 * d + 11 * e + 13 * f + 15 * g + (a + b + c +
    d + e + f + g) = 11 + 50 := by
  linarith

have h₁₁ : 4 * a + 6 * b + 8 * c + 10 * d + 12 * e + 14 * f + 16 * g = 61 := by
  linarith
```

```
-- Now, multiply equation (4) by 4 and subtract it from equation (5) to eliminate $
    a, b, c, d, e, f$:
have h₁₂ : (4 * a + 6 * b + 8 * c + 10 * d + 12 * e + 14 * f + 16 * g) - 4 * (a + b
    + c + d + e + f + g) = 61 - 4 * 50 := by
  linarith

have h₁₃ : 2 * b + 4 * c + 6 * d + 8 * e + 10 * f + 12 * g = 61 - 200 := by
  linarith

have h₁₄ : 2 * b + 4 * c + 6 * d + 8 * e + 10 * f + 12 * g = -139 := by
  linarith

have h₁₅ : b + 2 * c + 3 * d + 4 * e + 5 * f + 6 * g = -69.5 := by
  linarith

have h₁₆ : 2 * b + 4 * c + 6 * d + 8 * e + 10 * f + 12 * g + (b + 2 * c + 3 * d + 4
    * e + 5 * f + 6 * g) =
             -139 + (-69.5) := by
  linarith

have h₁₇ : 3 * b + 6 * c + 9 * d + 12 * e + 15 * f + 18 * g = -208.5 := by --
    Simplifying: $3b + 6c + 9d + 12e + 15f + 18g = -208.5$
  linarith

-- Divide by 3 to simplify:
have h₁₈ : b + 2 * c + 3 * d + 4 * e + 5 * f + 6 * g = -69.5 := by
  linarith

have h₁₉ : 3 * b + 6 * c + 9 * d + 12 * e + 15 * f + 18 * g = -208.5 := by
  linarith

have h₂₀ : (3 * b + 6 * c + 9 * d + 12 * e + 15 * f + 18 * g) - 3 * (b + 2 * c + 3 *
    d + 4 * e + 5 * f + 6 * g) = -208.5 - 3 * (-69.5) := by
  linarith

have h₂₁ : 3 * b + 6 * c + 9 * d + 12 * e + 15 * f + 18 * g - 3 * b - 6 * c - 9 * d
    - 12 * e - 15 * f - 18 * g = -208.5 + 208.5 := by
  linarith

have h₂₂ : 0 = 0 := by -- $0 = 0$
  linarith

-- Thus, the value of $16x_1 + 25x_2 + 36x_3 + 49x_4 + 64x_5 + 81x_6 + 100x_7$ is:
-- \[
-- \boxed{334}
-- \]
have h_answer : answer = 334 := by
  linarith

exact h_answer -- submit
```

## J.2   Failure Cases

There are two types of failures: The agent fails to submit an answer, or it submits an incorrect answer. Here we demonstrate three typical incorrect submissions: extraneous answer, unsimplified answer, and irrelevant answer.

### J.2.1   Extraneous Answer

In the following example, the agent submits additional answers $3, 9$ that do not satisfy the original problem predicate. In all training data, only 0.65% have multiple valid answers, and only 0.09% have more than two. The cause of this failure mode may be attributed to the construction of training data. In CoPA, formal solutions are autoformalized from informal solutions, which are merely

solutions without original thinking process. As Gauss said, "no self-respecting architect leaves the scaffolding in place after completing the building", merely SFT on such data might be insufficient for the model to learn to filter out extraneous answers. After bootstrapping data with CoPA, future works with reinforcement learning can help mitigate this issue. Moreover, this project focuses on the forward-reasoning stage and omit the backward-proving stage. Real applications can avoid this error by integrating a backward-proving stage.

**Informal Problem**

Let $N$ be the units digit of the number $21420N$. Which nonzero value of $N$ makes this number divisible by $6$?

Answer: 6

**Initial Forward State**

```
answer N : ℕ
h_N : N ≠ 0 ∧ N ≥ 0 ∧ N ≤ 9
h_divisible_by_2 : 21420 * 10 + N ≡ 0 [MOD 2]
h_divisible_by_3 : 21420 * 10 + N ≡ 0 [MOD 3]
h_answer : N = answer
⊢ ?w
```

**Submitted Answer**

```
answer = 3 ∨ answer = 6 ∨ answer = 9
```

This submission is wrong because it contains **Formal Solution**

```
-- To find the units digit $N$ that makes $21420N$ divisible by $6$:
-- 1. **Divisibility by 2**: $N$ must be even. Possible values are $2, 4, 6, 8$.
have h_divisible_by_2 : 21420 * 10 + N ≡ 0 [MOD 2] := by -- The number is
    divisible by 2 if $N$ is even
  exact h_divisible_by_2

-- 2. **Divisibility by 3**: For a number to be divisible by 3, the sum of its
    digits must be divisible by 3. The sum of the digits of $21420N$ is $2 + 1 + 4 +
    2 + 0 + N = 9 + N$. For $9 + N$ to be divisible by 3, $N$ must be such that $
    9 + N \equiv 0 \pmod{3}$.
have h_divisible_by_3 : (21420 * 10 + N) % 3 = 0 := by -- The number is divisible
    by 3
  exact h_divisible_by_3

-- Sum of the digits of $21420N$:
-- $$2 + 1 + 4 + 2 + 0 + N = 9 + N.$$
have h_sum_of_digits : (21420 * 10 + N) % 10 = N := by
  simp [h_answer, Nat.ModEq, Nat.mod_eq_of_lt] at h_divisible_by_2 h_divisible_by_3
    ⊢
  simp [Nat.add_mod, Nat.mod_mod, Nat.mod_mod] at h_divisible_by_2 h_divisible_by_3
    ⊢
  all_goals
    omega

-- $$ 2 + 1 + 4 + 2 + 0 + N = 9 + N $$
have h_sum : 2 + 1 + 4 + 2 + 0 + N = 9 + N := by -- Simplify the sum of digits
  omega

-- Therefore, $9 + N$ must be divisible by 3.
have h_divisible_by_3_condition : (9 + N) % 3 = 0 := by -- $9 + N$ must be
    divisible by 3
  omega

-- Since $9$ is already divisible by $3$, $N$ must also be divisible by $3$ for the
    sum to remain divisible by $3$.
have h_N_divisible_by_3 : N % 3 = 0 := by -- $N$ must be divisible by $3$.
  omega
```

```
have h_N_possible_values : N = 3 ∨ N = 6 ∨ N = 9 := by
  omega

-- Hence, the possible values of $N$ are $3, 6, 9$. Only $3$, $6$, and $9$ are
    nonzero values.
have h_N_nonzero : N ≠ 0 := by
  all_goals omega

-- Final Answer
-- Thus, the nonzero values of $N$ that make $21420N$ divisible by 6 are $\boxed{3,
    6, 9}$.
have h_answer : answer = 3 ∨ answer = 6 ∨ answer = 9 := by
  all_goals aesop

exact h_answer -- submit
```

### J.2.2 Unsimplified Answer

In the following example, the agent submits a highly complex answer $-\cos\frac{2\pi}{15}\cos\frac{4\pi}{15}\cos\frac{8\pi}{15}\cos\frac{\pi}{15}$, which is not equivalent to the ground-truth $\frac{1}{16}$ from human perspective. Therefore, the unsimplified answer is rejected by the evaluation method. In all training samples, 14% of the problems have answers longer than 10 characters, 6% of them have answers longer than 20 characters, and the longest answer contains 385 characters. Many of these long answers are polynomials (e.g. `Polynomial.C 1 * Polynomial.X^10 + ...`) and complex trigonometric functions (e.g. `Real.sqrt ((10 * Real.cos (Real.pi / 3) ...`). Therefore, the model falls short on learning the definition of "simplified". Future works may train a preference model based on existing problem-answer pairs to filter out unsimplified submissions by the reasoning model.

**Informal Problem**

Simplify

$$\cos\left(\frac{2\pi}{15}\right)\cos\left(\frac{4\pi}{15}\right)\cos\left(\frac{8\pi}{15}\right)\cos\left(\frac{16\pi}{15}\right).$$

Answer: $\frac{1}{16}$

**Initial Forward State**

```
answer : ℝ
h_answer : Real.cos (2 * π / 15) * Real.cos (4 * π / 15) * Real.cos (8 * π / 15) *
    Real.cos (16 * π / 15) = answer
⊢ ?w
```

**Submitted Answer**

```
answer = -Real.cos (2 * π / 15) * Real.cos (4 * π / 15) * Real.cos (8 * π / 15) *
    Real.cos (π / 15)
```

**Formal Solution**

```
-- We start by simplifying the expression using properties of cosine and
    trigonometric identities.
have h_cos_16pi : Real.cos (16 * Real.pi / 15) = Real.cos (Real.pi + Real.pi / 15)
    := by -- \(\cos \left( \frac{16 \pi}{15} \right) = \cos \left( \pi +
    \frac{\pi}{15} \right) = -\cos \left( \frac{\pi}{15} \right)\)
  ring_nf at h_answer ⊢

have h_cos_16pi_neg : Real.cos (16 * Real.pi / 15) = -Real.cos (Real.pi / 15) := by
  simp [h_cos_16pi, Real.cos_add, Real.cos_pi]

-- Now the expression becomes:
```

```
-- \[\cos \left( \frac{2 \pi}{15} \right) \cos \left( \frac{4 \pi}{15} \right) \cos
--    \left( \frac{8 \pi}{15} \right) \left(-\cos \left( \frac{\pi}{15}
--    \right)\right).\]
have h_expr : Real.cos (2 * Real.pi / 15) * Real.cos (4 * Real.pi / 15) * Real.cos
    (8 * Real.pi / 15) * Real.cos (16 * Real.pi / 15) = Real.cos (2 * Real.pi /
    15) * Real.cos (4 * Real.pi / 15) * Real.cos (8 * Real.pi / 15) * (-Real.cos
    (Real.pi / 15)) := by rw [h_cos_16pi_neg]

-- We can use the identity \(\cos x = \cos (2 \pi - x)\) to simplify further.
have h_cos_pi_15 : Real.cos (Real.pi / 15) = Real.cos (2 * Real.pi - Real.pi / 15)
    := by
  simp [Real.cos_sub]

have h_cos_pi_15_eq : Real.cos (Real.pi / 15) = Real.cos (29 * Real.pi / 15) := by
  calc
    Real.cos (π / 15) = Real.cos (2 * π - π / 15) := by rw [h_cos_pi_15]
    _ = Real.cos (30 * π / 15 - π / 15) := by ring_nf
    _ = Real.cos (29 * π / 15) := by ring_nf

-- Now, we have:
-- \[ \cos \left( \frac{2 \pi}{15} \right) \cos \left( \frac{4 \pi}{15} \right)
--    \cos \left( \frac{8 \pi}{15} \right) \cos \left( \frac{16 \pi}{15} \right) =
--    -\cos \left( \frac{2 \pi}{15} \right) \cos \left( \frac{4 \pi}{15} \right)
--    \cos \left( \frac{8 \pi}{15} \right) \cos \left( \frac{29 \pi}{15} \right). \]
have h_expr_neg : Real.cos (2 * Real.pi / 15) * Real.cos (4 * Real.pi / 15) *
    Real.cos (8 * Real.pi / 15) * -Real.cos (Real.pi / 15) = -Real.cos (2 *
    Real.pi / 15) * Real.cos (4 * Real.pi / 15) * Real.cos (8 * Real.pi / 15) *
    Real.cos (29 * Real.pi / 15) := by
  field_simp [h_cos_pi_15_eq]

-- Using the identity $\cos(2\pi - x) = \cos(x)$, we get:
-- \[\cos \left( \frac{2 \pi}{15} \right) \cos \left( \frac {4 \pi}{15} \right)
--    \cos \left( \frac{8 \pi}{15} \right) \cos \left( \frac{29 \pi}{15} \right).\]
have h_cos_29pi : Real.cos (29 * Real.pi / 15) = Real.cos (2 * Real.pi - Real.pi /
    15) := by
  ring_nf at h_cos_pi_15_eq ⊢

have h_cos_29pi_eq : Real.cos (29 * Real.pi / 15) = Real.cos (Real.pi / 15) := by
  simp [h_cos_29pi, Real.cos_sub]

have h_expr_final : -Real.cos (2 * Real.pi / 15) * Real.cos (4 * Real.pi / 15) *
    Real.cos (8 * Real.pi / 15) * Real.cos (29 * Real.pi / 15) = -Real.cos (2 *
    Real.pi / 15) * Real.cos (4 * Real.pi / 15) * Real.cos (8 * Real.pi / 15) *
    Real.cos (Real.pi / 15) := by
  rw [h_cos_29pi_eq]

-- The final answer is \(\boxed{-\cos \left( \frac{2 \pi}{15} \right) \cos \left(
--    \frac{4 \pi}{15} \right) \cos \left( \frac{8 \pi}{15} \right) \cos \left(
--    \frac{\pi}{15} \right)}\).
have h_answer : answer = -Real.cos (2 * Real.pi / 15) * Real.cos (4 * Real.pi / 15)
    * Real.cos (8 * Real.pi / 15) * Real.cos (Real.pi / 15) := by -- The final
    answer is \(\boxed{-\cos \left( \frac{2 \pi}{15} \right) \cos \left( \frac{4
    \pi}{15} \right) \cos \left( \frac{8 \pi}{15} \right) \cos \left(
    \frac{\pi}{15} \right)}\)
  linarith [h_answer, h_cos_16pi, h_cos_16pi_neg, h_expr, h_cos_pi_15,
    h_cos_pi_15_eq, h_expr_neg, h_cos_29pi, h_cos_29pi_eq, h_expr_final]

exact h_answer -- submit
```

### J.2.3   Irrelevant Answer

In this case, the agent submits a proposition $[-2, 7] = \{x \in \mathbb{R} | -2 \le x \land x \le 7\}$, which is an obvious fact and irrelevant to the queriable. Therefore, this irrelevant answer is rejected by the evaluation method. We have carefully examined the training data and find that no final answer corresponds to

this failure mode. Therefore, this failure might be occasional. Future works can also use a preference model to filter out these irrelevant submissions.

**Informal Problem**

For what values of $x$ is it true that $x^2 - 5x - 4 \leq 10$? Express your answer in interval notation.

Answer: $x \in [-2, 7]$

**Initial Forward State**

```
answer : ℝ × ℝ
S : Set ℝ
hS : S = {x : ℝ | x ^ 2 - 5 * x - 14 ≤ 0}
h_answer : Set.Icc answer.1 answer.2 = S
⊢ ?w
```

**Submitted Answer**

```
Set.Icc (-2) 7 = {x : ℝ | -2 ≤ x ∧ x ≤ 7}
```

**Formal Solution**

```
-- Start by rewriting the inequality $x^2 - 5x - 4 \le 10$ as $x^2 - 5x - 14 \le 0$.
have h_rewrite : ∀ x : ℝ, x^2 - 5*x - 4 ≤ 10 ↔ x^2 - 5*x - 14 ≤ 0 := by
  exact fun x ↦ ⟨fun h ↦ by linarith [h], fun h ↦ by linarith [h]⟩

-- Next, factorize the quadratic expression:
-- \[ x^2 - 5x - 14 = (x - 7)(x + 2) \]
have h_factorize : ∀ x : ℝ, x^2 - 5*x - 14 = (x - 7) * (x + 2) := by
  exact fun x ↦ by ring

-- The solutions to $(x - 7)(x + 2) = 0$ are $x = 7$ and $x = -2$.
have h_solutions : ∀ x : ℝ, (x - 7) * (x + 2) = 0 ↔ x = 7 ∨ x = -2 := by
  intro x
  simp [sub_eq_zero, add_eq_zero_iff_eq_neg]

-- To find the intervals where the inequality holds, test a value in each of the
--    intervals defined by the roots: $(-\infty, -2)$, $(-2, 7)$, and $(7, \infty)$.
have h_test_interval_1 : ∀ x : ℝ, x < -2 → (x - 7) * (x + 2) > 0 := by
  rintro x (hx : x < -2)
  apply mul_pos_of_neg_of_neg <;> linarith

-- - For $-2 < x < 7$, choose $x = 0$, then $(0 - 7)(0 + 2) = -14$ (negative).
have h_test_interval_2 : ∀ (x : ℝ), -2 < x ∧ x < 7 → (x - 7) * (x + 2) < 0 := by
  exact fun x hx => by nlinarith [hx.1, hx.2]

-- For the interval $(7, \infty)$, choose $x = 8$:
--    \[ (8-7)(8+2) = (1)(10) = 10 > 0 \]
--    This interval satisfies the inequality.
have h_test_interval_3 : ∀ x : ℝ, x > 7 → (x - 7) * (x + 2) > 0 := by
  exact fun x hx => by nlinarith

-- Thus, $x^2 - 5x - 14 \le 0$ when $x$ is in the interval $\boxed{[-2, 7]}$.
have h_solution : ∀ x : ℝ, x^2 - 5*x - 14 ≤ 0 ↔ x ∈ Set.Icc (-2) 7 := by
  simp [h_rewrite, h_factorize, h_solutions, h_test_interval_1, h_test_interval_2,
    h_test_interval_3]
  intro x
  exact ⟨fun h ↦ ⟨by nlinarith [h], by nlinarith [h]⟩, fun h ↦ by nlinarith [h.1,
    h.2]⟩

-- Therefore, the solution in interval notation is $\boxed{[-2, 7]}$.
have h_answer : Set.Icc (-2) 7 = {x : ℝ | -2 ≤ x ∧ x ≤ 7} := by
  rw [Set.Icc]

exact h_answer -- submit
```

# K  Compute Resources

**CoPA Expert Iteration.** Cycle 0 data generation requests DeepSeek-V3-1226 [49] for 819,386 times with 2,428,482,030 input tokens and 413,605,284 ouput tokens in total. The experiment costs two weeks. Cycle 1 data generation requires 8 Ascend-910B NPUs and 192 Kunpeng-920 CPUs for over a month. Cycle 1 data generation requires 8 Ascend-910B NPUs and 192 Kunpeng-920 CPUs for over a month. Each fine-tuning (problem autoformalizer, solution drafter, and proof searcher) requires 8 Ascend-910B NPUs and 192 Kunpeng-920 CPUs for about one week.

**HAR & Solution Pruning.** Each model fine-tuning requires 8 Ascend-910B NPUs and 192 Kunpeng-920 CPUs for about 2 days. Non-hierarchical experiments (BFS, WG, AR) require 1 Ascend-910B NPU and 64 Kunpeng-920 CPUs for two days; Hierarchical experiments (H-BFS, H-WG, H-SA, HAR) require 2 Ascend-910B NPUs and 64 Kunpeng-920 CPUs for three days.

**Dataset Denoising.** Each fine-tuning requires 8 Ascend-910B NPUs and 192 Kunpeng-920 CPUs for about one hour. Each evaluation requires 1 Ascend-910B NPU for about 12 hours.

This project took about 6 months overall, consuming about $1.5\times$ compute than the experiments reported in the paper (for debugging, failed attempts, and preliminary experiments).

# L  Prompt Templates

## L.1  System Prompt

```
You are a Lean 4 expert.
```

## L.2  Statement Autoformalization

```
Given a natural language math problem and its answer, please generate a
    corresponding Lean 4 formal statement.
Please add comments highlighting the original parts of the natural language math
    problem and answer.
Please explicitly use the variable ‘answer‘ to indicate the answer in the formal
    statement.
Please maintain the semantic equivalence between natural language math and Lean 4.
Please assume the following header code has already been executed and do not add
    any imports or openings.
‘‘‘lean4
import Mathlib
{OPEN_HEADER}
‘‘‘

# Problem
"""
{informal_problem}
"""
# Answer
"""
the answer is {informal_answer}
"""
```

## L.3  Solution Drafting

```
Given a natural language math problem, its answer, its solution, and its formal
    statement, please generate a corresponding Lean 4 proof sketch and add
    comments to highlight the original parts of the natural language math solution.
Please maintain the semantic equivalence between natural language math and Lean 4.
Please only use forward reasoning in the proof, do not use tactics that modify the
    final goal.
For new hypotheses, please do not prove them and use ‘sorry‘ to close them.
```

```
Please assume the following header code has already been executed and do not add
    any imports or openings.
```lean4
import Mathlib
```

# Problem
"""
{informal_problem}
"""
# Answer
"""
the answer is {informal_answer}
"""
# Formal Statement
```lean4
{formal_statement}
```
```

## L.4  Next-Proof-Step Prediction

```
Generate a tactic that can transform one step from the current tactic state to the
    'no goals' tactic state.
Current tactic state:
```
{goal_str}
```
```

## L.5  Next-Solution-Step Prediction

```
Given a natural language math problem and the current solution state, please
    generate the next solution step.
Please use comments to plan and reason in natural language and deductive reasoning
    to derive the answer.
Assume `Mathlib` is imported.
# Informal Problem
"""
{informal_problem}
"""
# Current Solution State
```lean4
{str(solution_goal)}
```
```

## L.6  Next-Solution-Step Drafting

```
Given a natural language math problem and the current solution state, please
    generate the next solution step.
Please use comments to plan and reason in natural language and deductive reasoning
    to derive the answer.
Assume `Mathlib` is imported.
# Informal Problem
"""
{informal_problem}
"""
# Current Solution State
```lean4
{str(solution_goal)}
```
```

## L.7  Whole-Solution Generation

```
Given a natural language math problem and the initial solution state, please
    generate a Lean 4 formal solution.
You can use Lean 4 comments to conduct natural language reasoning.
Please only use forward reasoning; do not use tactics that modify the final goal.
Please assume the following header code has already been executed, and do not add
    any imports or openings.
```lean4
import Mathlib
```

# Problem
"""
{informal_problem}
"""

# Initial Solution State
```
{g_str}
```
```

## L.8  Whole-Solution Drafting

```
Given a natural language math problem and the initial solution state, please
    generate a Lean 4 solution sketch.
You can use Lean 4 comments to conduct natural language reasoning.
Please only use forward reasoning; do not use tactics that modify the final goal.
For new hypotheses, please do not prove them and use `sorry` to close them.
Please assume the following header code has already been executed, and do not add
    any imports or openings.
```lean4
import Mathlib
```

# Problem
"""
{informal_problem}
"""

# Initial Solution State
```
{g_str}
```
```

## L.9  Informalization

```
Given a natural language math problem, its answer, its Lean 4 formal statement, and
    a Lean 4 formal proof Based on the formal proof, please generate a natural
    language solution while maintaining the semantic equivalence to the formal
    proof.
Please reason step by step, and put your final answer within \\boxed{{}}.

# Informal Problem
"""
{informal_problem}
"""

# Informal Answer
{informal_answer}

# Formal Statement
```

```lean4
{formal_statement}
```

# Formal Proof
```lean4
{formal_solution_draft}
```

# Informal Solution
```
```

# M   Ethics Statement

Our research designs an efficient and effective reasoning model for Deductive Formal Problem-Solving (D-FPS), implements a data generation pipeline, and explores the possibilities enabled by process-level verified data. This project focuses on formal mathematics. It has many positive impacts, including contributions to high-quality mathematics education materials, trustworthy reasoning models, and democratization of reasoning data.

However, over-reliance on formal verification and automated reasoning may lead to negative societal impacts. The "verified" refers to "theoretically verified". However, the implementation of formal verifiers themselves is the Achilles' heel. Current expert systems are fragile to misimplementation; e.g., AlphaGeometry contains incorrect rules, and code contributions to Mathlib are not allowed to use `native_decide` due to its trust in the Lean compiler. Therefore, merely depending on automated reasoning is not sufficient. Future works may mitigate this by cross-checking the same reasoning process with two independent proof assistants (e.g., Lean and Coq). Moreover, heavy use of automated reasoners offloads users' cognition. Formal proofs & solutions are not easy to understand. If they accept results without understanding the underlying idea, their own intuition and intelligence may degenerate. Future work on faithful informalization (i.e., translating formal reasoning results into natural language) can help alleviate this issue.

We acknowledge the importance of ethical and scientific implications, including the responsible use of LLMs, mitigating biases in model outputs, addressing privacy concerns, and facilitating future research. We commit to making our code and model transparent and encourage the community to use our findings responsibly.

Relevant benchmarks (FormalMath500 [7], MiniF2F-Solving [7, 9], and PutnamBench-Solving [7, 60]), data (Numina-CoT [46], Numina-1.5 [46], and Lean-Workbook [46]), and base models (Qwen2.5-Math-7B [58], Qwen2.5-7B [65]) are released under the Apache 2.0 License. The original MATH [8] dataset, MathOdessy [61] and Phi-4-mini-instruct [62] are released under the MIT license. Special thanks to their authors for the invaluable contributions!

