# OpenReview forum: "Bootstrapping Hierarchical Autoregressive Formal Reasoner with Chain-of-Proxy-Autoformalization"
_NeurIPS.cc/2025/Conference — NeurIPS 2025 poster_

### Official Review · Reviewer_ZCQb · 2025-06-29

**Clarity:** 3
**Significance:** 3
**Originality:** 3
**Rating:** 3
**Confidence:** 4

**Summary:**

This paper introduces a dual-component system to advance Deductive Formal Problem-Solving (D-FPS). The first component, the Hierarchical Autoregressive Formal Reasoner (HAR), is a novel reasoning pipeline that aims to improve efficiency and align with human problem-solving by decomposing each reasoning step into a high-level, informal-aligned "drafting" phase and a low-level, formal "proving" phase. It constructs solutions autoregressively, with per-step feedback to minimize wasted effort. The second component, the Chain-of-Proxy-Autoformalization (CoPA) pipeline, addresses the critical scarcity of formal training data. It generates process-verified formal problem-solution pairs from informal text by chaining together tasks like statement autoformalization and proof drafting, refining its models through an expert iteration loop. The authors report that HAR achieves state-of-the-art results on key benchmarks (FormalMath500, MiniF2F-Solving) and that CoPA is a scalable data generation method, further exploring applications in solution pruning and dataset denoising.

**Questions:**

1.	Could you refine the definition of "backtracking" used in the paper? Given that the agent "repeatedly retries" a failed step, how do you draw the line between this "per-step rejective sampling" and a localized form of backtracking? A more precise terminology would strengthen the paper's claims.

2.	How can the paper claim there is "no statistically significant difference" in the solution pruning experiment (Table 4) while also stating in the checklist that no statistical tests were performed? This is a direct contradiction. Could you either provide the statistical analysis that supports this claim or rephrase the claim as an observation pending formal statistical validation?

3.	Could you explain the scenario in Table 2 where HAR consumes a higher budget than H-WG for FormalMath500? This seems to contradict the general efficiency claim. What factors in HAR's process could lead to higher budget consumption in certain cases compared to its hierarchical variants?

4.	Could you clarify the intent behind comparing HAR's problem-solving success rate with H-SA's solution autoformalization rate? Since H-SA is given the ground-truth informal solution, how does this comparison fairly evaluate HAR's end-to-end reasoning capability rather than just its final output?

5.	The current evaluation is minimal. It should include other math benchmarsks such as PutnamBench, MathOdyssey, ISARSTEP, etc. It should also be expanded to include other reasoning datasets such as Folio, Proofwriter, PrOntoQA, AR-LSAT, etc. Will this also work with other open weight models or is it targeted only for one model? Some evaluation here would also be desirable.

**Ethical Concerns:**

["NO or VERY MINOR ethics concerns only"]

**Final Justification:**

Thank you for adding more results and I hope you incorporate them appropriately in the paper. Based on this I have updated my score.

**Limitations:**

The authors provide a "Limitations and Future Works" section in the appendix. However, it fails to address the most critical limitations of the work. It mentions the desire for "more expert iteration" and extension to other domains but completely ignores the huge compute cost. A proper limitations section must prominently discuss this issue, as it is the primary factor that determines the practical relevance and reproducibility of this research. The current limitations section reads more like a "future work" proposal and does not engage in a discussion of the work's constraints.

**Quality:**

2

**Strengths And Weaknesses:**

Strengths:

•	The dual proposals of HAR and CoPA represent a comprehensive and ambitious attempt to build a complete system for formal reasoning. The HAR architecture, which combines hierarchical decomposition with autoregressive generation, appears to be a new pipeline design. Similarly, CoPA's approach of chaining proxy tasks to bootstrap data is a new strategy for tackling the data bottleneck.

•	The headline results are impressive. HAR shows a dramatic improvement in solving rates over previous methods, jumping from 15.50% to 43.39% on FormalMath500 and from 21.87% to 55.68% on MiniF2F-Solving. The ablation studies effectively demonstrate the value of the hierarchical component, with H-WG, for example, improving the performance of WG significantly.

•	The appendices detail the environment, training hyperparameters, prompt templates, and evaluation settings.


Weaknesses:

•	The paper claims HAR operates "without backtracking." However, the method's description of "repeatedly retry until the budget is exhausted" upon failure of a step is functionally a form of local backtracking or re-sampling. While different from global proof-search backtracking, the terminology is imprecise and potentially misleading.

•	A weakness is the claim that solution pruning results in "no statistically significant difference" in performance, while the authors simultaneously state in the checklist that no statistical significance tests were run due to computational cost. Making a statistical claim without providing statistical evidence is a methodological flaw.

•	The abstract's claim that HAR achieves its results with a "lower budget" is not universally supported by the data. In Table 2, HAR consumes a higher budget than its hierarchical counterpart H-WG on the FormalMath500 benchmark (Cycle 1). This inconsistency weakens the general claim of superior efficiency across all comparisons.

•	The comparison between HAR and H-SA (a method using ground-truth informal solutions) is misleading. H-SA's primary task is solution formalization, not end-to-end problem-solving from scratch. Framing HAR's victory as outperforming a ground-truth-guided method overstates the case, as they are not performing the same task.

•	The claim of "consistent scalability" for CoPA is complicated by the data in Table 1, which shows a significant drop in the "Gap" filling success rate from Cycle 1 to Cycle 2. While explained by data filtering, this suggests that scaling all components of the pipeline in unison presents a challenge.

•	The current evaluation is minimal. It should include other math benchmarsks such as PutnamBench, MathOdyssey, ISARSTEP, etc. It should also be expanded to include other reasoning datasets such as Folio, Proofwriter, PrOntoQA, AR-LSAT, etc. Will this also work with other open weight models or is it targeted only for one model? Some evaluation here would also be desirable.


Quality:

The paper presents an innovative approach. The core architectures of HAR and CoPA are well-conceived, and the overall system demonstrates good empirical power. The detailed appendices point to a high degree of rigor in the implementation. However, the quality is impacted by several issues in the reporting and analysis: imprecise terminology, making statistical claims without statistical tests, limited dataset evaluations, and comparisons that could be framed more carefully.

Clarity:

The paper is generally well-written and organized but tries to stuff too much into the limited page limit that one must constantly go to the appendix which should not be the case. The motivations for both HAR and CoPA are well-articulated, and the use of figures is effective in illustrating the core pipelines.

Significance:

The work is significant in its problem domain. It offers a potential path forward on the major challenges of verifiability and data scarcity in automated reasoning. If the practical issues (like compute cost) can be mitigated, the proposed architectures could have a substantial impact on the development of more robust and trustworthy AI systems. However, the immense computational cost described in the appendix may limit the work's immediate practical impact for the broader research community.

Originality:

The conceptual framework is the paper's strongest point. The core ideas of hierarchical reasoning to bridge the informal-formal granularity gap and chaining proxy tasks to bootstrap data are original and insightful. While built upon existing concepts, the synthesis and application within the D-FPS context are novel. The paper introduces new ways of thinking about both reasoning architecture and data generation for formal mathematics.

---

> ### Author Rebuttal · Authors · 2025-07-31
>
> We greatly appreciate your detailed reviews, as well as your constructive comments:) We plan to update (U) the following points in the camera-ready version:
>
> - U1. Add a footnote in line 30 "backtracking," to avoid misunderstanding.
> - U2. Temporarily rewrite line 297-298 as "demonstrates marginal differences".
> - U3. Re-run main experiments and solution pruning experiments twice to report Avg and StDev.
> - U4. Visualize solving rate - budget curve Sec. 5.2.
> - U5. Add discussion [R4] in Sec. 5.1.
> - U6. Replace the claim "consistent scalability" with "scalability".
> - U7. Add PutnamBench-Solving results in Table 2.
>
> We sincerely hope them to improve the paper quality and satisfactorily respond to your comments. Our detailed responses (R) are as follows.
>
> > W1, Q1. Refining the definition of "backtracking".
>
> R1. Thank you for this suggestion. In this paper, "backtracking" refers to the global proof-search backtracking. We will add a footnote at its first occurrence (line 30) to avoid readers' confusion.
>
> > W2, Q2. Improper claim of "no statistically significant difference".
>
> R2. We will temporarily rewrite our claim (line 297-298) as "demonstrates marginal differences" and schedule two repeated runs on solution pruning experiments (Table 4) and main experiments (Table 2. Cycle 1: BFS, WG, AR, H-BFS, H-AR; Cycle 2: H-AR). Due to the limited response time and computing resources, we have not finished all re-runs yet. We promise to update Table 2, Table 4, and line 297-298 in the camera-ready version.
>
> Some of them are finished and show strong statistical significance.
>
> | Benchmark | FormalMath500 |  | | MiniF2F-Solving |  |  |
> | --------- | ------------- | -------------- | ----------------- | --------------- | -------------- | ------------------ |
> | Method  | All | Solved%  | Steps | All | Solved%  | Steps  |
> | BFS | 378 | 9.52% ± 0.57%  | 28138.67 ± 104.32 | 370 | 9.64% ± 1.66%  | 27711.67 ± 338.99  |
> | WG  | 378 | 18.78% ± 0.22% | 36935.00 ± 44.14  | 370 | 24.95% ± 1.09% | 42339.67 ± 1212.55 |
> | AR  | 378 | 34.39% ± 0.78% | 20860.00 ± 233.20 | 370 | 44.41% ± 0.34% | 17814.33 ± 139.76  |
> | H-WG  | 378 | 36.77% ± 0.86% | 17038.67 ± 461.32 | |  |  |
>
> > W3, Q3. Explanation of Table 2 (budgets of H-WG and HAR)
>
> R3. We have visualized the solving rate - budget curve of SFT-based experiments in Table 2, which demonstrates that HAR is nearly Pareto-optimal (except for extremely low budget scenarios). Due to the response policy, we provide an interpolated table here. The visualization will be added to Sec. 5.1 in the camera-ready version.
>
> | Cycle | Benchmark | FormalMath500 |  |  |  | MiniF2F-Solving |  |  |  |
> | ----- | --------------- | ------------- | ---------- | ---------- | ---------- | --------------- | ---------- | ---------- | ---------- |
> | | Method \ Budget | 3000  | 5000 | 10000  | 20000  | 3000  | 5000 | 10000  | 20000  |
> | 1 | BFS | 0.52% | 1.93%  | 8.20%  | 9.79%  | 0.54% | 1.23%  | 5.80%  | 10.81% |
> | 1 | WG  | 4.76% | 9.58%  | 17.46% | 18.52% | 3.30% | 11.62% | 22.60% | 23.78% |
> | 1 | AR  | 22.86%  | 33.50% | 34.13% | 34.13% | 35.29%  | 43.51% | 44.86% | 44.86% |
> | 1 | H-BFS | 0.78% | 3.44%  | 8.20%  | 14.29% | 0.60% | 2.21%  | 6.40%  | 12.70% |
> | 1 | H-WG  | **29.01%**  | 36.77% | 36.77% | 36.77% | **38.23%**  | 45.41% | 45.95% | 45.95% |
> | 1 | HAR | 21.27%  | **40.50%** | **42.59%** | **42.59%** | 30.36%  | **52.26%** | **54.86%** | **54.86%** |
> | 2 | HAR | 32.33%  | 42.06% | 43.39% | 43.39% | 38.81%  | 54.07% | 55.68% | 55.68% |
>
> Moreover, we count the budget distribution of 4 types of problems: _Both Succeeded_ (both HAR and H-WG successfully solve the problem), _One Succeeded_ (only one of them successfully solves), and _Both Failed_ (both of them fail to solve). The budget consumption of each type is as below:
>
> | Step Count | Both Succeeded (131) | One Succeeded (38) | Both Failed (209) | Sum |
> | ---------- | -------------------- | ------------------ | ----------------- | ----- |
> | HAR | 1236 | 959  | 16720 | 18915 |
> | H-WG | 933  | 1967 | 13574 | 16474 |
>
> As in Appendix F.1, we set the number of expansion attempts as $K=80$ for BFS&AR, and the number of generation attempts as $K=8$ for WG. This is because in the dataset constructed by CoPA, each formal solution contains 9.7 steps on average (line 708).
>
> In "Both Failed": H-WG generates 8.12 steps per attempt, i.e., 64.95 steps per problem. HAR explores at most 80 steps per problem. Therefore, the budget consumption of HAR seems to be higher than H-WG.
>
> > W4, Q4. Explanation of the comparison between HAR and H-SA.
>
> R4. H-SA is an augmented version of H-WG. H-WG generates a solution sketch given the initial solution state, and then fills the proof gaps; H-SA generates a solution sketch given both the formal statement (can be viewed as initial solution state + ground-truth formal answer) and the ground-truth informal solution, then fills the proof gaps.
>
> In this view, H-SA sets an (intuitive, not rigorous) upper bound of H-WG. Now that HAR outperforms H-SA, we have a better chance of believing that HAR consistently outperforms H-WG.
>
> We will add the above explanation of the indent behind, including H-SA for comparison in Sec. 5.1.
>
> > W5. Explanation of CoPA's scalability in Table 1.
>
> R5. We agree with your insight that scaling up all components of CoPA faces a challenge. The informal dataset is fixed as Numina-1.5. Therefore, as CoPA iterates from formalizing samples and removing successful ones from future iterations, the remaining data will be more and more difficult, until 1) CoPA fails on all of them, or 2) CoPA succeeds on all of them. However, as stated in Appendix B, we need more expert iterations to determine CoPA's "consistent scalability" or "inscalability".
>
> Thank you for this comment. We will remove the claim of "consistent scalability" in the abstract.
>
> > W6, Q5. More evaluation.
>
> R6. We have evaluated methods in Table 2 (excluding H-SA due to the absence of ground-truth informal solution) on the PutnamBench-Solving[1,2], which consists of 324 problems from undergraduate-level competitions. The results are as follows.
>
> | Cycle | Experiment | Solved |
> | ----- | ---------- | ------ |
> | ~ | ICL  | 0  |
> | ~ | Hybrid CoT | 0  |
> | 1 | BFS  | 1  |
> | 1 | WG | 0  |
> | 1 | AR | 2  |
> | 1 | H-BFS  | 1  |
> | 1 | H-WG | 2  |
> | 1 | HAR  | 2  |
> | 2 | HAR  | 2  |
>
> Although AR / HAR demonstrates superiority over baselines, all methods almost fail on this benchmark because of its out-of-distribution (OOD). As stated in Appendix B. "Limitation", CoPA formalizes Numina-CoT and Numina-1.5, which focuses on "Chinese high school math exercises to US and international mathematics olympiad competition problems". PutnamBench-Solving focuses on undergrad-level competitions and is totally OOD from the training set. The OOD curse in formal reasoning is more challenging since each formal step should be not only semantically but syntactically correct. LLM usually fails to call functions or apply theorems without prior knowledge. As mentioned in line 599-601, we admit current data domain is not enough and encourage future work on this.
>
> Other benchmarks, such as MathOdyssey, ISARSTEP, are not suitable for our evaluation. This project focuses on Lean 4 language, proposing CoPA that translates informal problem-solution pairs into formal ones, and HAR that searches for a formal solution given a formal problem. They are unsuitable for informal math QA (MathOdyssey), filling in a missing intermediate proposition given surrounding proofs (ISARSTEP), natural language inference (FOLIO, ProofWriter, PrOntoQA), and natural language QA (AR-LSAT).
>
> Among them, HAR can be indirectly evaluated on MathOdyssey by 1) autoformalizing the informal problems and answers; 2) conducting D-FPS using HAR. However, this evaluation will be greatly affected by the accuracy of autoformalization. If you feel it necessary to conduct such an evaluation, we will do it ASAP.
>
> > L1. Limitation on computing cost
>
> R7. CoPA's data flow (problem autoformalization, solution drafting, gap filling) does not result in a higher-order computational complexity compared with autoformalization-based approaches in theorem proving (statement autoformalization, proof search). Moreover, CoPA differs from [3] by:
> 1. Problem/Statement Autoformalization: CoPA uses heuristic filtering instead of LLM grader (Appendix C), therefore requires fewer LLM calls while having higher accuracy.
> 2. Solution/Proof Generation: CoPA uses H-SA, which demonstrates a significantly higher success rate and lower budget cost than WG, as demonstrated in Table 2.
>
> HAR shows superior efficiency compared to baseline methods (R3). Therefore, the theoretical computational resource requirement is not higher than other methods.
>
> The seemingly high compute cost in Sec. J is because this project is conducted on specific hardware platforms that often lack open-source support and in-depth optimization. We believe researchers equipped with A100s, H100s, or even B100s can reproduce the pipeline with much lower resources.
>
> [1] Liu, Qi, et al. "Beyond Theorem Proving: Formulation, Framework and Benchmark for Formal Problem-Solving." arXiv preprint arXiv:2505.04528 (2025).
>
> [2] Tsoukalas, George, et al. "Putnambench: Evaluating neural theorem-provers on the putnam mathematical competition." Advances in Neural Information Processing Systems 37 (2024): 11545-11569.
>
> [3] Lin, Yong, et al. "Goedel-prover: A frontier model for open-source automated theorem proving." arXiv preprint arXiv:2502.07640 (2025).

---

> > ### Comment · Reviewer_ZCQb · 2025-08-05
> >
> > Thank you to the authors for running a few more experiments and updating tables 2 and 4 (in the next update R2). However, the current evaluation is minimal as stated in Question 5. In its current form, I do not think the paper is ready for publication - it needs significantly more evaluation. The explanation for Q3 is also not convincing. So, I will keep my rating as is.

---

> ### Author Response · Authors · 2025-08-04
> **Update: Tables 2 & 4 [R2]**
>
> We have now completed three independent runs and will update Tables 2 & 4 with means ± standard deviations.
>
> Table 2 (Main Results) confirms the robustness of our findings:
>
> | Cycle | Method | FormalMath500 | | | MiniF2F-Solving | | |
> | --- | --- | --- | --- | --- | --- | --- | --- |
> | | | Solved | Budget | Avg. Len | Solved | Budget | Avg. Len |
> | 1 | BFS | 9.52% ± 0.57% | 28139 ± 104 | 4.31 ± 0.08 | 9.64% ± 1.66% | 27712 ± 339 | 4.24 ± 0.17 |
> | 1 | WG | 18.78% ± 0.22% | 36935 ± 44 | 14.24 ± 0.79 | 24.95% ± 1.09% | 42340 ± 1213 | 16.03 ± 0.49 |
> | 1 | AR | 34.39% ± 0.78% | 20860 ± 233 | 7.66 ± 0.18 | 44.41% ± 0.34% | 17814 ± 140 | 8.12 ± 0.33 |
> | 1 | H-BFS | 13.32% ± 1.47% | 27777 ± 360 | 4.71 ± 0.11 | 12.25% ± 1.47% | 27479 ± 203 | 4.64 ± 0.22 |
> | 1 | H-WG | 36.77% ± 0.86% | 17039 ± 461 | 7.10 ± 0.08 | 47.30% ± 1.01% | 16303 ± 388 | 7.37 ± 0.10 |
> | 1 | HAR | 43.65% ± 1.35% | 18432 ± 344 | 7.92 ± 0.11 | 55.50% ± 0.46% | 15069 ± 143 | 8.96 ± 0.07 |
> | 2 | HAR | 44.09% ± 0.54% | 18273 ± 116 | 7.81 ± 0.30 | 56.58% ± 0.92% | 14754 ± 269 | 8.74 ± 0.23 |
>
> Table 4 (Solution Pruning) shows a more interesting pattern.
>
> | Recursion | FormalMath500 | | | MiniF2F-Solving | | |
> | --- | --- | --- | --- | --- | --- | --- |
> | | Solved | Budget | Avg. Len | Solved | Budget | Avg. Len |
> | 0 | 34.39% ± 0.78% | 20860 ± 233 | 7.66 ± 0.18 | 44.41% ± 0.34% | 17814 ± 140 | 8.12 ± 0.33 |
> | 1 | 34.30% ± 0.54% | 20785 ± 148 | 7.00 ± 0.01 | 44.59% ± 0.66% | 17646 ± 133 | 7.34 ± 0.22 |
> | 2 | 34.57% ± 0.87% | 20737 ± 232 | 7.07 ± 0.21 | 44.50% ± 0.13% | 17633 ± 44 | 7.19 ± 0.05 |
> | 3 | 34.83% ± 0.33% | 20625 ± 74 | 6.88 ± 0.30 | 45.50% ± 1.04% | 17425 ± 258 | 7.63 ± 0.11 |
> | ∞ | 34.48% ± 0.76% | 20709 ± 207 | 6.68 ± 0.16 | 45.32% ± 0.13% | 17468 ± 66 | 7.49 ± 0.20 |
>
> **FormalMath500**: There is no significant change in solving rates across pruning recursions, and average solution lengths decrease monotonically.
>
> **MiniF2F-Solving**: For $R=0,1,2$, solving rates remain relatively stable and average solution lengths decrease monotonically. For $R=3, \infty$, solving rates improve and the average solution lengths fluctuate.
>
> We hypothesize this is due to **limited context length** (as discussed in [R1] to Reviewer pJF9) on complicated problems. Deeper pruning improves reasoning efficiency, preventing models from exceeding the context window (4096 tokens).
>
> Let $S_0, S_\infty$ denote problems solved in $R=0, \infty$ experiments, respectively. The mean solution length of $S_0 \backslash S_\infty$ is $9.82$, while that of $S_\infty \backslash S_0$ is $11.85$. The longest solution in $S_0 \backslash S_\infty$ has 17 steps, while $S_\infty \backslash S_0$ contains three solutions longer than 17 steps (19, 24, 29).
>
> This further supports the benefits of solution pruning: training efficiency (Train Size $\downarrow$) and reasoning efficiency (Avg. Len. $\downarrow$, Solved $\uparrow$).
>
> We will also add the above analysis in Appendix G.

---

> ### Author Response · Authors · 2025-08-07
> **Response to Reviewer ZCQb (1/2)**
>
> We appreciate the opportunity for further clarification :)
>
> > Explanation for Q3 is also not convincing.
>
> R8. Thank you for this insightful comment. We have re-examined the code, data, and experiment results, and discovered a bug in WG's step-counting script: it double-counted every proof step that belonged to a solution step.
>
> After correcting it, we reran WG and H-WG using 16 generation attempts ($K_W=16$). The updated Table 2 is shown below.
>
> | Table 2 (BudgetUpdated) | Benchmark | FormalMath500 | | MiniF2F-Solving | |
> | --- | --- | --- | --- | --- | --- |
> | Cycle | Method | Solved | Budget | Solved | Budget |
> | 1 | BFS | 9.52% ± 0.57% | 28139 ± 104 | 9.64% ± 1.66% | 27712 ± 339 |
> | 1 | **WG ($K_W=8$)** | **18.78% ± 0.22%** | **18456 ± 92** | **24.95% ± 1.09%** | **18853 ± 454** |
> | 1 | **WG ($K_W=16$)** | **21.78% ± 0.12%** | **35391 ± 153** | **28.83% ± 0.77%** | **35895 ± 487** |
> | 1 | AR | 34.39% ± 0.78% | 20860 ± 233 | 44.41% ± 0.34% | 17814 ± 140 |
> | 1 | H-BFS | 13.32% ± 1.47% | 27777 ± 360 | 12.25% ± 1.47% | 27479 ± 203 |
> | 1 | H-WG ($K_W=8$) | 36.77% ± 0.86% | 17039 ± 461 | 47.30% ± 1.01% | 16303 ± 388 |
> | 1 | **H-WG ($K_W=16$)** | **40.04% ± 0.50%** | **32134 ± 226** | **51.08% ± 0.44%** | **30357 ± 212** |
> | 1 | HAR | 43.65% ± 1.35% | 18432 ± 344 | 55.50% ± 0.46% | 15069 ± 143 |
> | 2 | HAR | 44.09% ± 0.54% | 18273 ± 116 | 56.58% ± 0.92% | 14754 ± 269 |
>
> **Comparisons of (BFS, WG ($K_W=16$), AR) and (H-BFS, H-WG ($K_W=16$), HAR) demonstrate the superiority of AR/HAR.**
>
> To understand the remaining budget gap on FormalMath500, we broke down the budget consumption of H-WG ($K_W=8$) and HAR, where "Both Failed" denotes problems that both methods fail to solve, and "Others" denotes problems that any method solves.
>
> | Benchmark | Method | Both Failed | Others | Sum |
> | --- | --- | --- | --- | --- |
> | FormalMath500 | #Problem | 209 | 169 | 378 |
> | | Budget HWG | 13574 | 2900 | 16474 |
> | | Budget HAR | 16720 | 2195 | 18915 |
> | | Budget Difference | **+3146** | **-705** | **+2441** |
> | MiniF2F-Solving | #Problem | 161 | 209 | 370 |
> | | Budget HWG | 11800 | 3960 | 15760 |
> | | Budget HAR | 12880 | 2475 | 15355 |
> | | Budget Difference | **+1080** | **-1485** | **-405** |
>
> For the 209 "Both Failed" problems in FormalMath500, **H-WG generates 65 steps (8.1 steps per solution)**, while HAR exhausted its full budget of 80 steps. On MiniF2F-Solving, H-WG's step count (9.2 steps per solution) is closer to CoPA's dataset average of 9.7 (line 708), so the gap is smaller.
>
> Overall, as demonstrated by the **visualizations of solving rate - budget curve [R3]**, AR/HAR remains more efficient than WG/H-WG when both are given comparable generation limits.
>
> We will include the corrected numbers and the new visualization in the camera-ready. Thank you for this constructive comment, which really helps us to improve this work :)

---

> ### Author Response · Authors · 2025-08-07
> **Response to Reviewer ZCQb (2/2)**
>
> > Q5. The current evaluation is minimal. It should **include other math benchmarks** such as PutnamBench, MathOdyssey, ISARSTEP, etc. It should also be **expanded to include other reasoning datasets** such as Folio, Proofwriter, PrOntoQA, AR-LSAT, etc. **Will this also work with other open weight models** or is it targeted only for one model? Some evaluation here would also be desirable.
>
> R9. We appreciate your suggestions and have respectfully done in [R10, R11]. Due to time and resource constraints, some evaluations have not been completed yet. We will continue and update in the camera-ready.
>
> We respectfully note that, for D-FPS, the current evaluation covers all publicly available benchmarks (FormalMath500, MiniF2F-Solving, and PutnamBench-Solving). Sec. 4.2.4 proposes _dataset denoising_, which expands to improving data quality of informal math reasoning. Further expansions are valuable but lie outside this paper’s focus.
>
> > Including informal math benchmarks MathOdyssey.
>
> R10. Expanding to informal benchmarks such as MathOdyssey implies a novel task: _end-to-end formal problem-solving (E2E-FPS)_ (like [1]) that chains (i) autoformalizing a natural language problem, and (ii) generating a formal solution. This has sparked a long discussion among authors.
>
> We provide a pilot study using CoPA Autoformalizer to formalize 288 problems with non-empty labels in MathOdessy, resulting in 269 successfully formalized problems. Models are evaluated with environmental & hyperparameter settings identical to Appendices D & F. We also evaluated AR/HAR with $K_S = 40$ for fair comparison with WG/H-WG $K_W=8$ (WG/H-WG $K_W=16$ takes longer time).
>
> | Qwen2.5 (BudgetUpdated) | MathOdessy | |
> | --- | --- | --- |
> | Experiment | Solved | Budget |
> | BFS | 11.90% | 19490 |
> | WG | 15.61% | 12773 |
> | **AR ($K_S=40$)** | **24.16%** | **8613** |
> | AR | 24.16% | 16773 |
> | H-BFS | 15.24% | 19233 |
> | H-WG | 26.39% | 12672 |
> | **HAR ($K_S=40$)** | **27.14%** | **8417** |
> | HAR | 28.25% | 16156 |
>
> Comparison between AR/HAR ($K_S=40$) and baselines **validate AR/HAR's superiority over BFS/H-BFS and WG/H-WG**. We will continue $K_W=16$ experiments and incorporate the discussions into a new appendix.
>
> > Effectiveness on other open weight models.
>
> R11. Sincere apologies for the late response, as these experiments are expensive and time-consuming. Honestly, we too have wondered whether the impressive effectiveness is a flash in the pan on Qwen series models (as recent pushback on the unreasonable effectiveness of Qwen-based RL[2]).
>
> We have SFTed `Phi-4-mini-instruct` with identical recipes in the main experiments (Cycle 1 data; Appendices E & K). We also evaluated AR/HAR with $K_S = 40$ for fair comparison with WG/H-WG $K_W=8$ (WG/H-WG $K_W=16$ takes longer time). Early results are shown below.
>
> | Phi-4 (BudgetUpdated) | Benchmark | FormalMath500 | | MiniF2F-Solving | | Putnam-Solving | | MathOdessy | |
> | --- | --- | --- | --- | --- | --- | --- | --- | --- | --- |
> | Cycle | Method | Solved | Budget | Solved | Budget | Solved | Budget | Solved | Budget |
> | 1 | BFS | 7.41% | 28665 | 6.76% | 28230 | 0.31% | 25857 | 12.64% | 19408 |
> | 1 | WG | 14.02% | 20295 | 21.08% | 20768 | 0.00% | 13987 | 13.38% | 14026 |
> | **1** | **AR ($K_S=40$)** | **25.66%** | **12198** | **31.35%** | **11560** | **0.62%** | **12893** | **24.54%** | **8688** |
> | 1 | AR | 26.72% | 23338 | 33.24% | 21464 | 0.62% | 25773 | 24.54% | 16808 |
> | 1 | H-BFS | 11.90% | 27992 | | | | | | |
> | 1 | H-WG | 33.07% | 17457 | | | | | | |
> | **1** | **HAR ($K_S=40$)** | **41.80%** | **10196** | | | | | | |
> | 1 | HAR | 42.06% | 18956 | | | | | | |
>
>
> **Results demonstrate similar pattern to `Qwen2.5-Math-7B` (Table 2; Updated [R2]; [R6]) and align well with our main claims.** On all 4 benchmarks, AR ($K_S=40$) demonstrates consistent pareto-optimality over BFS and WG. On FormalMath500, HAR ($K_S=40$) demonstrates similar superiority over H-BFS and H-WG.
>
> We will incorporate all new results and discussion in the camera-ready.
>
> ---
>
> Thank you again for your constructive suggestions, which have made our work more complete :)
>
>
> [1] Mathesis: Towards Formal Theorem Proving from Natural Languages
>
> [2] Spurious Rewards: Rethinking Training Signals in RLVR

---

> > ### Author Response · Authors · 2025-08-08
> > **Update: Qwen2.5 (MathOdessy) and Phi-4 (All)**
> >
> > We have now completed the SFT of Phi-4 with Cycle 2 data, as well as the evaluations of Cycle 1 WG/H-WG with $K_W=16$ and Cycle 2 HAR. The updated results are shown in the tables below.
> >
> > | Qwen2.5 (BudgetUpdated) | | MathOdessy | |
> > | --- | --- | --- | --- |
> > | Cycle | Experiment | Solved | Budget |
> > | 1 | BFS | 11.90% | 19490 |
> > | 1 | WG | 15.61% | 12773 |
> > | 1 | WG $(K_W=16)$ | 18.22% | 24865 |
> > | 1 | **AR $(K_S=40)$** | **24.16%** | **8613** |
> > | 1 | **AR** | **24.16%** | **16773** |
> > | 1 | H-BFS | 15.24% | 19233 |
> > | 1 | H-WG | 26.39% | 12672 |
> > | 1 | H-WG $(K_W=16)$ | 28.62% | 24074 |
> > | 1 | **HAR $(K_S=40)$** | **27.14%** | **8417** |
> > | 1 | **HAR** | **28.25%** | **16156** |
> > | 2 | **HAR $(K_S=40)$** | **27.88%** | **8377** |
> > | 2 | **HAR** | **29.00%** | **16038** |
> >
> > | Phi-4 (BudgetUpdated) | Benchmark | FormalMath500 | | MiniF2F-Solving | | Putnam-Solving | | MathOdessy | |
> > | --- | --- | --- | --- | --- | --- | --- | --- | --- | --- |
> > | Cycle | Method | Solved | Budget | Solved | Budget | Solved | Budget | Solved | Budget |
> > | 1 | BFS | 7.41% | 28665 | 6.76% | 28230 | 0.31% | 25857 | 12.64% | 19408 |
> > | 1 | WG | 14.02% | 20295 | 21.08% | 20768 | 0.00% | 13987 | 13.38% | 14026 |
> > | 1 | WG $(K_W=16)$ | 15.61% | 40153 | 25.41% | 39366 | 0.00% | 27973 | 15.24% | 27356 |
> > | **1** | **AR $(K_S=40)$** | **25.66%** | **12198** | **31.35%** | **11560** | **0.62%** | **12893** | **24.54%** | **8688** |
> > | **1** | **AR** | **26.72%** | **23338** | **33.24%** | **21464** | **0.62%** | **25773** | **24.54%** | **16808** |
> > | 1 | H-BFS | 11.90% | 27992 | 11.35% | 27591 | 0.31% | 25844 | 11.90% | 19679 |
> > | 1 | H-WG | 33.07% | 17457 | 41.35% | 17306 | 0.00% | 14306 | 25.28% | 12911 |
> > | 1 | H-WG $(K_W=16)$ | 38.10% | 33693 | 47.03% | 33742 | 0.31% | 30658 | 25.65% | 23894 |
> > | **1** | **HAR $(K_S=40)$** | **41.80%** | **10196** | **49.19%** | **9247** | | | **27.88%** | **8328** |
> > | **1** | **HAR** | **42.06%** | **18956** | **49.73%** | **16687** | | | **27.88%** | **16088** |
> > | **2** | **HAR  $(K_S=40)$** | **42.33%** | **10247** | | | | | | |
> > | **2** | **HAR** | **42.86%** | **18902** | | | | | | |
> >
> > The new results across multiple base models (Qwen2.5 and Phi-4) and benchmarks (FormalMath500, MiniF2F-Solving, Putnam-Solving, and MathOdessy) consistently demonstrate the effectiveness and generalizability of our proposed methods. The only exception is observed in the Qwen2.5 experiments on MathOdessy, where Cycle 1 H-WG ($K_W=16$) slightly outperforms Cycle 1 HAR ($28.62\\%$ vs. $28.25\\%$) at the cost of $1.5\times$ budget.
> >
> > We are grateful for your valuable comments and the opportunity to further validate our claims. We will incorporate the above discussion and updates into the camera-ready version. If any remaining concerns prevent a higher score, we would be happy to provide further clarification.

---

> > > ### Comment · Reviewer_ZCQb · 2025-08-08
> > >
> > > Thank you and I really appreciate the efforts by the authors in engaging and providing a lot more results. Incorporating all the new results will require significant effort in the updated paper and it is not clear if the clarity will suffer greatly as we do not get to see the updates. I would suggest you revise the paper with some more results that you are still working on and make it a much stronger paper. However, I will raise my score by a point, given the current form.

---

> > > > ### Author Response · Authors · 2025-08-09
> > > >
> > > > We sincerely appreciate your kind feedback and are grateful for raising the score. We’ll complete the remaining experiments, streamline the presentation, and ensure the camera-ready version is both clear and strengthened. Your suggestions have been invaluable in helping us improve this paper :)

---

### Official Review · Reviewer_pJF9 · 2025-06-30

**Clarity:** 3
**Significance:** 3
**Originality:** 3
**Rating:** 4
**Confidence:** 3

**Summary:**

This paper studies the problem of formal mathematical problem solving, specifically using Lean to write formal proofs for mathematical problems. The paper proposes a LLM-based proof generation method, called HAR.  One core idea of HAR is to decompose proofs and align the granularity between informal and formal reasoning at more atomic steps, while restricting backtracking to improve efficiency. To support model training, the paper also introduces an approach for auto-formalizing informal problems and solutions into formal proofs. The paper evaluates HAR on two datasets. The results show strong improvements over baseline methods in both performance and computational efficiency.

**Questions:**

Related to test-time scaling, how would you enable sampling with HAR? Would this involve allowing multiple paths at each proof state?

**Ethical Concerns:**

["NO or VERY MINOR ethics concerns only"]

**Final Justification:**

I thank authors for the responses and would like to keep my score on the positive side.

**Limitations:**

yes

**Quality:**

3

**Strengths And Weaknesses:**

## Strengths

The paper presents a thorough investigation of the formal proof generation pipeline, covering both data generation  and the proof search methodology.

The proposed HAR method achieves significant strong gains over baselines while maintaining computational efficiency.

The paper includes auto-formalized datasets that could benefit the broader research community working on formal mathematics and theorem proving.

The paper is generally well-written and easy to understand


## Weakness

The proposed approach restricts proof generation into an autoregressive way without back tracking.  I have some concerns about test-time scaling capabilities. While this design choice improves efficiency, it may constrain the model's ability to explore complex proof strategies that indeed require backtracking. I wonder if we allow sufficient budget for different approaches (such as allowing more samples), what the results would look like compared to baselines and other systems in the literature.


While the paper shows strong improvements over chosen baselines, I feel it could benefit from  strong automated provers in the literature, such as Godel-prover [1] and STP [2], which  also leverage auto-formalized training data and archives strong performance on Mini-F2F.  acknowledging different experimental settings (models, compute resources), While the settings (e.g., models and training compute) can be different, the paper could benefit from more comparison and discussion with respect to related work.


[1] Goedel-Prover: A Frontier Model for Open-Source Automated Theorem Proving

[2] STP: Self-play LLM Theorem Provers with Iterative Conjecturing and Proving

---

> ### Author Rebuttal · Authors · 2025-07-31
>
> Thank you for the appreciation of our work and the inspiring suggestions. We sincerely hope the planned paper updates (U) and responses \(R\) can help resolve your concerns.
>
> U1. Add test-time scaling experiments and discussions[R1] as a new Appendix.
>
> U2. Polish Sec. 2 and add discussions[R2]
>
> U3. Polish caption of Fig. 1 to reflect discussion[R3]
>
> > W1. Impact of backtracking on test-time scaling capabilities.
>
> R1. We set the max step limit $K=320$ to evaluate the test-time scalability of Cycle 2 H-BFS and HAR on the FormalMath500 benchmark. HAR solves 47.88% problems with 64727 search steps, while H-BFS solves 32.54% with 90533 steps. The following table compares the two methods with interpolated solving rates.
>
> | Budget | 2500   | 5000   | 10000  | 20000  | 40000  | 60000  | 80000  | 90533 |
> | ------ | ------ | ------ | ------ | ------ | ------ | ------ | ------ | ------ |
> | H-BFS  | 0.43%  | 2.81%  | 9.10%  | 16.32% | 27.25% | 31.22% | 32.54% | 32.54% |
> | HAR    | 24.10% | 44.39% | 47.35% | 47.88% | 47.88% | 47.88% | ~      |   ~    |
>
> H-BFS shows consistent scalability and plateaus after ~70000 steps. The solving rate of HAR rapidly increases and is saturated after ~20000 steps. The saturated solving rate of HAR (47.88%) is significantly higher than that of H-BFS (32.54%).
>
> We find that HAR plateaus because of limited context length. All models are SFTed from Qwen2.5-Math-7B, whose maximum context length is 4096 tokens. Complicated problems require longer CoTs, which require the model to have a larger context length to cover the solution state (more and more conclusions along reasoning) and next-step draft.
>
> HAR experiment encounters 953 times of context length exceeding (experiment restarts upon an exception occurs), while H-BFS encounters 413 times. This corresponds to our observation (line 286) that BFS with accumulated log-prob as value function tends to collapse to breadth-first search.
>
> We will add the above discussions as a new Appendix.
>
> Moreover, we have visualized the solving rate - budget curve of SFT-based experiments, which demonstrates that HAR is nearly pareto-optimal (except extreme low budget scenarios). Due to the response policy, we provide an interpolated table here. The visualization will be added to Sec.5.1 in the camera-ready version.
>
> | Cycle | Benchmark       | FormalMath500 |            |            |            | MiniF2F-Solving |            |            |            |
> | ----- | --------------- | ------------- | ---------- | ---------- | ---------- | --------------- | ---------- | ---------- | ---------- |
> |       | Method \ Budget | 3000          | 5000       | 10000      | 20000      | 3000            | 5000       | 10000      | 20000      |
> | 1     | BFS             | 0.52%         | 1.93%      | 8.20%      | 9.79%      | 0.54%           | 1.23%      | 5.80%      | 10.81%     |
> | 1     | WG              | 4.76%         | 9.58%      | 17.46%     | 18.52%     | 3.30%           | 11.62%     | 22.60%     | 23.78%     |
> | 1     | AR              | 22.86%        | 33.50%     | 34.13%     | 34.13%     | 35.29%          | 43.51%     | 44.86%     | 44.86%     |
> | 1     | H-BFS           | 0.78%         | 3.44%      | 8.20%      | 14.29%     | 0.60%           | 2.21%      | 6.40%      | 12.70%     |
> | 1     | H-WG            | 29.01%        | **36.77%** | 36.77%     | 36.77%     | **38.23%**      | 45.41%     | 45.95%     | 45.95%     |
> | 1     | HAR             | 21.27%        | 40.50%     | **42.59%** | **42.59%** | 30.36%          | **52.26%** | **54.86%** | **54.86%** |
> | 2     | HAR             | 32.33%        | 42.06%     | 43.39%     | 43.39%     | 38.81%          | 54.07%     | 55.68%     | 55.68%     |
>
> > W2. More comparison and discussion with respect to related work.
>
> R2. For HAR, Sec.5.1. "Baseline Methods" covers most existing methods, to name a few:
> - WG: DeepSeekProver-V1, DeepSeekProver-V1.5, DeepSeekProver-V2 (w/o Subgoal Decomposition), Goedel Prover, etc. belong to this category. They directly sample a whole proof given the formal statement.
> - H-WG: DeepSeekProver-V2 (w Subgoal Decomposition) belongs to this category. It firstly generates a formal proof sketch containing `sorry` placeholders, and generates proofs to fill these subgoals.
> - BFS: A wide spectrum of works following LeanDojo belongs to this category. They model the construction of proofs as a best-first search on a tree consisting of proof states and proof steps.
> - H-BFS: POETRY belongs to this category. It conducts best-first search while allowing the model to generate `sorry` placeholder and recursively search proofs for subgoals.
>
> HAR differs from existing methods by:
> - WG/H-WG: Splits the generation of formal solutions into many smaller rejective sampling steps to avoid error propagation.
> - BFS/H-BFS: Removing backtracking for more efficient D-FPS.
> - WG/BFS: Decoupling one formal step into an informal-aligned drafting and finer-grained gap filling.
>
> As for CoPA, it differs from [1] by:
> 1. Problem/Statement Autoformalization: CoPA uses heuristic filtering instead of LLM grader (Appendix C), therefore requires fewer LLM calls while having higher accuracy.
> 2. Solution/Proof Generation: CoPA uses H-SA, which demonstrates a significantly higher success rate and lower budget cost than WG, as demonstrated in Table 2.
>
> CoPA differs from [2] by:
> 1. Motivation. D-FPS faces extreme scarcity of formal problem-solution data. Therefore, CoPA targets to construct initial data for model training. Formal Theorem Proving (FTP) has abundant data (LeanWorkbook, Mathlib4), thus STP[2] is initialized by SFT on them (Sec.3.1 in [2]) and targets to generate more high-quality data.
> 2. Data: CoPA transforms informal problem-solution pairs into formal ones, while STP[2] generates new conjectures and their proofs.
> 3. Incorporation of informal language. CoPA aims at generating informal-formal aligned problems and solution drafts, thus integrating novel techniques including proxy autoformalization and hierarchical reasoning. STP[2] focuses on FTP and only works on formal language.
>
> > Q1. HAR's sampling strategy.
>
> R3. In the current design, for each state, HAR only samples one path. As shown in Fig. 1 and illustrated in line 131-137, given a solution state (i.e., proof state), HAR generates an informal thought, then a formal step draft, executes it, and attempts to fill the `sorry` gap with a low-level prover. This counts as 1 step in the budget computation. If the execution of gap filling fails, HAR generates another action until the budget is exhausted or the step is successfully executed and filled.
>
>
> [1] Lin, Yong, et al. "Goedel-prover: A frontier model for open-source automated theorem proving." arXiv preprint arXiv:2502.07640 (2025).
>
> [2] Dong, Kefan, and Tengyu Ma. "Stp: Self-play llm theorem provers with iterative conjecturing and proving." arXiv preprint arXiv:2502.00212 (2025).

---

> > ### Comment · Reviewer_pJF9 · 2025-08-06
> >
> > Thank you for your response! I appreciate the explanation with respect to related work (and I think it is good to include them in the revision). I would like to retain my score on the positive side.

---

> ### Author Response · Authors · 2025-08-07
>
> Thank you for the positive feedback! We're glad that our comprehensive pipeline and strong empirical gains resonated with you. We'll polish Sec. 2 "Related Works" and incorporate the comparisons in [R2].

---

### Official Review · Reviewer_6TF1 · 2025-07-03

**Clarity:** 3
**Significance:** 3
**Originality:** 3
**Rating:** 4
**Confidence:** 3

**Summary:**

This work addresses challenges in the domain of formal problem solving, aiming to resolve two key issues: (1) the misalignment between informal and formal reasoning granularity, and (2) the extreme scarcity of formal problem-solution pairs. To address the first issue, the authors propose a hierarchical autoregressive formal reasoner that decouples informal-aligned drafting from detailed formal proving. To tackle the second, they introduce Chain of Proxy Autoformalization—a data generation pipeline that cascades statement autoformalization, proof drafting, and proof search to simulate a proxy formalization path. Empirical evaluations demonstrate the effectiveness of the proposed approach.

**Questions:**

1. How does the runtime or computational efficiency compare with other baseline methods?

2. Can the proposed data generation pipeline produce highly challenging or competition-level problems?

**Ethical Concerns:**

["NO or VERY MINOR ethics concerns only"]

**Limitations:**

As noted in the Weaknesses section above.

**Quality:**

3

**Strengths And Weaknesses:**

Strengths:

1. The proposed method is well-motivated and addresses two important challenges with clear and coherent solutions.
2. The overall design is conceptually elegant and well-structured, with a clean division between drafting and proving.
3. The paper is well-written and easy to follow.
4. Experimental results are compelling and support the proposed contributions.

Weaknesses:

1. The approach requires considerable computational resources, which may hinder reproducibility or adoption by researchers without access to similar infrastructure.
2. Generalization remains a concern: While dataset denoising improves performance on most benchmarks, it underperforms on English multiple-choice and competition-style problems, indicating possible domain-specific limitations.

---

> ### Author Rebuttal · Authors · 2025-07-31
>
> Thank you for this insightful review and constructive comments. In summary, we will update the following points in the camera-ready version:
> - U1. Add discussions[R2] into Appendix H.
> - U2. Visualize solving rate - budget curve Sec. 5.2.
> - U3. Polish Sec.2 and add discussions[R3] (line 73)
> - U4. Polish Appendix B and add discussions [R4](line 598-599).
>
> Hope them can improve paper quality and help future readers. Our detailed responses (R ) are as follows.
>
> > W1. High compute resource requirement.
>
> R1. CoPA's data flow (problem autoformalization, solution drafting, gap filling) does not results in a higher-order computational complexity compared with autoformalization-based approaches in theorem proving (statement autoformalization, proof search). HAR show superior efficiency compared to baseline methods in Table 2. Therefore, the theoretical compute resource requirement is not higher than other methods.
>
> The seemingly high compute cost in Sec. J is because this project is conducted on specific hardware platforms that often lacks open-source support and in-depth optimization. We believe researchers equipped with A100s, H100s, or even B100s can reproduce the pipeline with much lower resource.
>
> As mentioned in NeurIPS Paper Checklist 5. We will open-source our code and model to facilitate future research.
>
> > W2. Dataset denoising's negative impact on specific datasets.
>
> R2. Dataset denoising falls behind on SVAMP, OlympiadBench, SAT-Math, MMLU-STEM, Gaokao-Math-QA, and AIME24, which may be due to:
> 1. **Multiple-choice Questions (MCQs) and True-false Problems (TFPs)** [SAT-Math, MMLU-STEM, Gaokao-Math-QA]. CoPA focuses on free-response problems (FRP), and therefore rewrites TFPs/MCQs into FRPs in Numina-CoT, Cycle 0, and filters out MCQs in Numina-1.5, Cycle n (line 197). This distribution shift might leave these problems out-of-distribution, thus negatively impacting the SFTed models' performance on benchmarks of MCQs, including SAT-Math, MMLU-STEM, and Gaokao-Math-QA.
> 2. **Simple math word problems (SMWPs)**. SVAMP focuses on SMWPs about simple arithmetic taught in grades four and lower. Highly automated tactics like `norm_num` and `linarith` can directly solve them, thus their formalization might collapse to the "guess-and-verify" pattern, or even only a simple `norm_num`. Then, the informalization might contain a less detailed CoT compared to the original solution.
> 3. **Out-of-distribution (OOD)**. STEM problems such as physics, biology, and astronomy are hard to formalize and OOD to Numina-1.5. For problems in geometry, probability, and combinatorics, their difficulty mainly lies in understanding the mathematical abstract and finding the expressions. Once formalized, most of them may collapse to simple arithmetics. This work focuses on problem-solving, therefore, theorem-proving ones are filtered out (line 183). Since MMLU-STEM consists of MCQs of STEM, OlympiadBench contains 25%+ physics problems and 21% about theorem-proving, which are relatively OOD to the SFT dataset and suffer from unstable performance. These limitations are included in Sec.B "Extension to Theorem Proving" and "Data Domain".
> 4. **Benchmark size**. AIME24 only contains 30 problems, and the evaluation pipeline (Qwen2.5-Math's official repo) uses greedy decoding with temperature $T=0$, sample number $n=1$. This setting is prone to randomness in small benchmarks like AIME24; one sample can result in 3.3% performance variation.
>
> The above analysis will be added to Appendix H in the camera-ready version.
>
> > Q1. Comparing computational efficiency with baselines.
>
> R3.
> - CoPA: To the best of our knowledge, this paper is the first to fully open all details about the computational cost. It is not easy to directly compare the computational cost of all components. Here we compare with [1] from a data flow perspective:
>     1. Problem/Statement Autoformalization: CoPA uses heuristic filtering instead of LLM grader (Appendix C), therefore requires fewer LLM calls while having higher accuracy.
>     2. Solution/Proof Generation: CoPA uses H-SA, which demonstrates a significantly higher success rate and lower budget cost than WG, as demonstrated in Table 2.
>
>     Therefore, CoPA demonstrates higher computational effciency. We will polish Sec.2. "Autoformalization" and integrate the above comparison in it.
> - HAR: Table 2 shows that HAR outperforms all baselines. Moreover, we have visualized the solving rate - budget curve of SFT-based experiments, which demonstrates that HAR is nearly pareto-optimal (except extreme low budget scenarios). Due to the response policy, we provide an interpolated table here. The visualization will be added to Sec.5.1 in the camera-ready version.
>
>
>
> | Cycle | Benchmark       | FormalMath500 |            |            |            | MiniF2F-Solving |            |            |            |
> | ----- | --------------- | ------------- | ---------- | ---------- | ---------- | --------------- | ---------- | ---------- | ---------- |
> |       | Method \ Budget | 3000          | 5000       | 10000      | 20000      | 3000            | 5000       | 10000      | 20000      |
> | 1     | BFS             | 0.52%         | 1.93%      | 8.20%      | 9.79%      | 0.54%           | 1.23%      | 5.80%      | 10.81%     |
> | 1     | WG              | 4.76%         | 9.58%      | 17.46%     | 18.52%     | 3.30%           | 11.62%     | 22.60%     | 23.78%     |
> | 1     | AR              | 22.86%        | 33.50%     | 34.13%     | 34.13%     | 35.29%          | 43.51%     | 44.86%     | 44.86%     |
> | 1     | H-BFS           | 0.78%         | 3.44%      | 8.20%      | 14.29%     | 0.60%           | 2.21%      | 6.40%      | 12.70%     |
> | 1     | H-WG            | **29.01%**        | 36.77% | 36.77%     | 36.77%     | **38.23%**      | 45.41%     | 45.95%     | 45.95%     |
> | 1     | HAR             | 21.27%        | **40.50%**     | **42.59%** | **42.59%** | 30.36%          | **52.26%** | **54.86%** | **54.86%** |
> | 2     | HAR             | 32.33%        | 42.06%     | 43.39%     | 43.39%     | 38.81%          | 54.07%     | 55.68%     | 55.68%     |
>
> > Q2. CoPA's generalizability to highly challenging or competition-level problems?
>
> R4. Yes. We believe that CoPA can better generalize to competition-level problems. Existing methods directly generates proofs, which may suffer from extreme sparsity on difficult problems. In contrast, CoPA conducts H-SA to utilize priors in natural language ground-truth and align reasoning granularity with informal reasoning.
>
> However, the current formalization system supports problems in disciplines including physics and combinatorics are not satisfactory. Recent progress in PhysLean[2] and CombiBench[3] can help future works to mitigate the gap.
>
> We will polish Appendix B (line 598-599) to integrate this discussion.
>
> [1] Lin, Yong, et al. "Goedel-prover: A frontier model for open-source automated theorem proving." arXiv preprint arXiv:2502.07640 (2025).
>
> [2] PhysLean: Digitalizing Physics in Lean 4
>
> [3] Liu, Junqi, et al. "CombiBench: Benchmarking LLM capability for combinatorial mathematics." arXiv preprint arXiv:2505.03171 (2025).

---

### Official Review · Reviewer_zBnd · 2025-07-05

**Clarity:** 2
**Significance:** 2
**Originality:** 2
**Rating:** 4
**Confidence:** 4

**Summary:**

The paper introduces two novel contributions to address challenges in Deductive Formal Problem-Solving (D-FPS) within formal theorem proving (FTP) environments. First, it proposes the Hierarchical Autoregressive Formal Reasoner (HAR), a reasoning pipeline that decouples high-level informal-aligned drafting from low-level formal proof searches, generating solution steps autoregressively without backtracking to enhance efficiency and alignment with informal reasoning. HAR achieves superior performance, solving 43.39% of FormalMath500 and 55.8% of MiniF2F-Solving datasets with minimal computational budget. Second, it presents Chain-of-Proxy-Autoformalization (CoPA), a data generation pipeline that leverages statement autoformalization, proof drafting, and proof search to create high-quality formal problem-solution pairs, addressing data scarcity. CoPA employs quality filtering (syntax, explosion, vacuous, and deductive checks) and expert iteration to improve data quality, demonstrating scalability through cycles of refinement. The paper also explores solution pruning and dataset denoising, validating their impact on efficiency and model performance. Comprehensive experiments demonstrate HAR’s effectiveness over baselines and CoPA’s scalability, with case studies illustrating successes and limitations.

**Questions:**

1. The case studies in Appendix I.2 highlight failures in trigonometric and divisibility problems. Could the authors provide a deeper analysis of these failure modes (e.g., specific limitations in HAR’s reasoning or CoPA’s data quality)? This could strengthen the paper’s quality score by demonstrating robustness or identifying clear areas for improvement.
2. The paper notes that repeated experiments for error bars are computationally expensive (Appendix J). Could the authors provide an alternative, such as confidence intervals based on existing runs or a subset of experiments, to support the reliability of solving rates?
3. Appendix L discusses positive impacts but lacks detail on potential negative societal implications (e.g., misuse of automated reasoning in critical domains). Could the authors elaborate on these risks and mitigation strategies?
4. How does HAR specifically improve upon prior methods like DeepSeek-Prover (references [24, 25]) in terms of technical innovation? A clearer differentiation could reinforce the originality score.
5. Table 8 shows mixed results for dataset denoising (e.g., performance drops on AQUA and ASUV). Could the authors clarify why denoising negatively affects certain datasets and how this impacts the method’s generalizability? This could improve the clarity score.

**Ethical Concerns:**

["NO or VERY MINOR ethics concerns only"]

**Limitations:**

Yes. The authors address limitations in Appendix I (failure cases) and Appendix L (societal impacts), noting the computational expense of statistical significance tests and the potential degradation of chain-of-thought reasoning in solution pruning.

**Paper Formatting Concerns:**

The paper adheres to NeurIPS 2025 formatting guidelines, with proper structure, references, and appendices.

**Quality:**

3

**Strengths And Weaknesses:**

**Strengths**
- The paper presents a technically robust approach with HAR and CoPA, supported by rigorous experimentation. HAR’s performance (43.39% on FormalMath500 and 55.8% on MiniF2F-Solving) significantly outperforms baselines (e.g., ICL, Hybrid CoT, BFS, WG) and even methods using ground-truth informal solutions (H-SA). CoPA’s expert iteration shows clear improvement, with problem autoformalization accuracy rising from 47% to 62% from Cycle 0 to Cycle 1. The use of Lean 4 for process-level supervision and the detailed evaluation on FormalMath500 and MiniF2F-Solving datasets strengthen the empirical claims. The inclusion of solution pruning and dataset denoising experiments adds depth to the analysis, showing practical efficiency gains.
- The paper is well-structured, with clear explanations of HAR and CoPA pipelines (e.g., Figures 1 and 2, Tables 1–8). The problem formulation as a Markov Decision Process (MDP) and the detailed descriptions of autoregressive search and quality filtering enhance understanding. Case studies in Appendix I provide concrete examples of successes and failures, aiding interpretability. The appendices (D, E, F, K) provide sufficient details on environment, training, inference, and prompt templates.

**Weaknesses**
- While the paper is generally clear, the mathematical notation for D-FPS problem formulation (Page 3) is dense and could benefit from more intuitive explanations or examples to make it accessible to readers unfamiliar with FTP. The discussion of dataset denoising (Appendix H, Table 8) is somewhat brief, making it hard to fully understand the impact on specific datasets (e.g., why performance drops on AQUA and ASUV).
- The paper’s claim of broad societal impact (Appendix L) is somewhat generic, focusing on positive impacts (e.g., mathematics education) without deeply exploring potential negative implications, such as over-reliance on automated reasoning in critical applications. The significance of solution pruning is also tempered by the observation that it may degrade chain-of-thought reasoning for simple problems (Page 19).

---

> ### Author Rebuttal · Authors · 2025-07-31
>
> We greatly appreciate your time and efforts in this detailed and invaluable review :) In the camera-ready version, we will update (U) each point:
> - U1. Rewrite Sec. 3 and Appendix A with example in Fig. 1 to explain FTP and D-FPS.
> - U2. Add discussions in R2 into Appendix H.
> - U3. Add discussions in R3 into Appendix L.
> - U4. Add more explanations in line 728.
> - U5. Add discussions in R5 into Appendix I.
> - U6. Update Table 2 with 3-run Avg and StDev.
> - U7. Polish Sec. 2 "Formal Reasoning".
>
> Hopefully, these updates can help improve the paper and satisfactorily respond to your comments. Our detailed responses (R) are as follows.
>
> > W1.1. Add intuitive explanations or examples for D-FPS.
>
> R1. We will further explain the mathematical notations for D-FPS and FTP in Sec. 3 and Appendix A (line 561-567) using the Fig. 1 example. Appendix A will be renamed as "More Discussions on D-FPS".
>
> > W1.2, Q5. Add more discussions of dataset denoising's negative impact on specific datasets.
>
> R2. Compared with _Direct SFT_, _Denoised SFT_ falls behind on SVAMP, OlympiadBench, SAT-Math, MMLU-STEM, Gaokao-Math-QA, and AIME24. The causes might be:
> 1. **Multiple-choice Questions (MCQs) and True-false Problems (TFPs)**. CoPA focuses on free-response problems (FRP), and therefore rewrites TFPs/MCQs into FRPs in Numina-CoT, Cycle 0, and filters out MCQs in Numina-1.5, Cycle n (line 197). This distribution shift might leave these problems out-of-distribution, thus negatively impacting the SFTed models' performance on benchmarks of MCQs, including SAT-Math, MMLU-STEM, and Gaokao-Math-QA.
> 2. **Simple math word problems (SMWPs)**. SVAMP focuses on SMWPs about simple arithmetic taught in grades four and lower[1]. Highly automated tactics like `norm_num` and `linarith` can directly solve them, thus their formalization might collapse to the "guess-and-verify" pattern, or even only a simple `norm_num`. Then, the informalization might contain a less detailed CoT compared to the original solution.
> 3. **Out-of-distribution (OOD)**. STEM problems, such as physics, biology, and astronomy, are hard to formalize and OOD to Numina-1.5. For problems in geometry, probability, and combinatorics, their difficulty mainly lies in understanding the mathematical abstract and finding the expressions. Once formalized, most of them may collapse to simple arithmetic. This work focuses on problem-solving, therefore, theorem-proving ones are filtered out (line 183). Since MMLU-STEM consists of MCQs of STEM, OlympiadBench contains 25%+ physics problems and 21% about theorem-proving, which are relatively OOD to the SFT dataset and suffer from unstable performance. These limitations are included in Sec. B "Extension to Theorem Proving" and "Data Domain".
> 4. **Benchmark Size**. AIME24 only contains 30 problems, and the evaluation pipeline (Qwen2.5-Math's official repo) uses greedy decoding with temperature $T=0$, sample number $n=1$. This setting is prone to randomness in small benchmarks like AIME24; one sample can result in 3.3% performance variation.
>
> > W2.1, Q3. Discussions about potential negative implications.
>
> R3. Thank you for this kind suggestion. We have explored more potential risks and mitigation strategies and will add them into Appenix L.
> 1. **Over-reliance on formal verification.** The "verified" refers to "theoretically verified". However, the implementation of formal verifiers themselves is the heel of Achilles. Current expert systems are fragile to misimplementation, e.g. Alphageometry contains wrong rules[2], code contributions to Mathlib are not allowed to use `native_decide` due to its trust on the Lean compiler. Therefore, merely depend on automated reasoning is not sufficient. Future works may mitigate this by cross-check the same reasoning process with two independent proof assistants (e.g. Lean and Coq).
> 2. **Over-reliance on automated reasoning.** Heavy use of automated reasoners offload users' covnitivity. Formal proofs / solutions are not easy to understand. If they accept results without understanding the idea behind, their own intuition and intelligence may degenerate. Future works about faithful informalization (translating formal reasoning results into natural language) can help alleviate this.
>
> > W2.2 Solution pruning degrade CoT for simple problems.
>
> R4. We acknowledge this comment and will add a suggestion in line 728 that encourage readers to apply solution pruning on difficult problems.
>
> > Q1. Deeper analysis of failure modes.
>
> R5. We have analyzed more examples and the distribution of our datasets, the results are as follows (and will be added into Appendix I).
> - **Extraneous Answer**: Only 0.65% of the formalized data have multiple valid answers, and only 0.09% have more than two. Formal solutions are autoformalized from informal solutions, which are merely solutions without original thinking process, as Gauss said, _"no self-respecting architect leaves the scaffolding in place after completing the building"_. Therefore, merely SFT on such data might be insufficient for the model to learn to filter out extraneous answers. After bootstrapping data with CoPA, future works with reinforcement learning can help mitigate this issue.
> - **Unsimplified Answer**: 14% of the problems have answers longer than 10 characters, 6% of them have answers longer than 20 characters, and the longest answer contains 385 characters. Many of these long answers are polynomials (e.g. `Polynomial.C 1 * Polynomial.X^10 + ...`) and complex trigonometric functions (e.g. `Real.sqrt ((10 * Real.cos (Real.pi / 3) ...`). Therefore, the model falls short on learning the definition of "simplified". Future works may train a preference model based on existing problem-answer pairs to filter out unsimplified submissions by the reasoning model.
> - **Irrelevant Answer**: We have carefully examined the training data and find that no final answer corresponds to this failure mode. Therefore, this failure might be occasional. Future works can also use a preference model to filter out these irrelevant submissions.
>
>
> > Q2. Statistical evidence to support the reliability of solving rates.
>
> R6. We will re-run main experiments (Table 2. Cycle 1: BFS, WG, AR, H-BFS, H-AR; Cycle 2: H-AR) for two more times, and report the average and standard deviation of total 3-runs. Due to the limited response time and compute resource, we have not finished all re-runs yet. We promise to update Table 2 in the camera-ready version. However, preliminary results below show strong statistical significance.
>
> | Benchmark | FormalMath500 |                |                   | MiniF2F-Solving |                |                    |
> | --------- | ------------- | -------------- | ----------------- | --------------- | -------------- | ------------------ |
> | Method    | All           | Solved%        | Steps             | All             | Solved%        | Steps              |
> | BFS       | 378           | 9.52% ± 0.57%  | 28138.67 ± 104.32 | 370             | 9.64% ± 1.66%  | 27711.67 ± 338.99  |
> | WG        | 378           | 18.78% ± 0.22% | 36935.00 ± 44.14  | 370             | 24.95% ± 1.09% | 42339.67 ± 1212.55 |
> | AR        | 378           | 34.39% ± 0.78% | 20860.00 ± 233.20 | 370             | 44.41% ± 0.34% | 17814.33 ± 139.76  |
> | H-WG      | 378           | 36.77% ± 0.86% | 17038.67 ± 461.32 |                 |                |                    |
>
> > Q4. Comparison between HAR and existing methods.
>
> R7. To respectively clarify, Sec.5.1. "Baseline Methods" covers most existing methods, to name a few:
> - WG: DeepSeekProver-V1[3], DeepSeekProver-V1.5[4], DeepSeekProver-V2[5] (w/o Subgoal Decomposition), Goedel Prover[6], etc. belongs to this category. They directly sample a whole proof given the formal statement.
> - H-WG: DeepSeekProver-V2[5] (Subgoal Decomposition) belongs to this category. It firstly generates a formal proof sketch containing `sorry` placeholders, and generates proofs to fill these subgoals.
> - BFS: A wide spectrum of works following LeanDojo belongs to this category. They model the construction of proofs as a best-first search on a tree consisting of proof states and proof steps.
> - H-BFS: POETRY belongs to this category. It conducts best-first search while allowing the model to generate `sorry` placeholder and recursively search proofs for subgoals.
>
> HAR differs from existing methods by
> - WG/H-WG: Splits the generation of formal solutions into many smaller rejective sampling steps to avoid error propagation.
> - BFS/H-BFS: Removing backtracking for more efficient D-FPS.
> - WG/BFS: Decoupling one formal step into an informal-aligned drafting and finer-grained gap filling.
>
> We will polish Sec.2 "Formal Reasoning" to differentiate HAR with existing methods.
>
>
> [1] Patel, Arkil, Satwik Bhattamishra, and Navin Goyal. "Are NLP models really able to solve simple math word problems?." arXiv preprint arXiv:2103.07191 (2021).
>
> [2] Lean Zulip Chat. "AlphaGeometry doesn't have a secure foundation"
>
> [3] Xin, Huajian, et al. "Deepseek-prover: Advancing theorem proving in llms through large-scale synthetic data." arXiv preprint arXiv:2405.14333 (2024).
>
> [4] Xin, Huajian, et al. "Deepseek-prover-v1. 5: Harnessing proof assistant feedback for reinforcement learning and monte-carlo tree search." arXiv preprint arXiv:2408.08152 (2024).
>
> [5] Ren, Z. Z., et al. "Deepseek-prover-v2: Advancing formal mathematical reasoning via reinforcement learning for subgoal decomposition." arXiv preprint arXiv:2504.21801 (2025).
>
> [6] Lin, Yong, et al. "Goedel-prover: A frontier model for open-source automated theorem proving." arXiv preprint arXiv:2502.07640 (2025).

---

> ### Author Response · Authors · 2025-08-04
> **Update: Table 2 [R6]**
>
> We have now completed three independent runs and will update Table 2 (and Table 4 for [R2] to Reviewer ZCQb) with the mean and standard deviation across runs. The results confirm the robustness of our findings:
>
> | Cycle | Method | FormalMath500 | | | MiniF2F-Solving | | |
> | --- | --- | --- | --- | --- | --- | --- | --- |
> | | | Solved | Budget | Avg. Len | Solved | Budget | Avg. Len |
> | 1 | BFS | 9.52% ± 0.57% | 28139 ± 104 | 4.31 ± 0.08 | 9.64% ± 1.66% | 27712 ± 339 | 4.24 ± 0.17 |
> | 1 | WG | 18.78% ± 0.22% | 36935 ± 44 | 14.24 ± 0.79 | 24.95% ± 1.09% | 42340 ± 1213 | 16.03 ± 0.49 |
> | 1 | AR | 34.39% ± 0.78% | 20860 ± 233 | 7.66 ± 0.18 | 44.41% ± 0.34% | 17814 ± 140 | 8.12 ± 0.33 |
> | 1 | H-BFS | 13.32% ± 1.47% | 27777 ± 360 | 4.71 ± 0.11 | 12.25% ± 1.47% | 27479 ± 203 | 4.64 ± 0.22 |
> | 1 | H-WG | 36.77% ± 0.86% | 17039 ± 461 | 7.10 ± 0.08 | 47.30% ± 1.01% | 16303 ± 388 | 7.37 ± 0.10 |
> | 1 | HAR | 43.65% ± 1.35% | 18432 ± 344 | 7.92 ± 0.11 | 55.50% ± 0.46% | 15069 ± 143 | 8.96 ± 0.07 |
> | 2 | HAR | 44.09% ± 0.54% | 18273 ± 116 | 7.81 ± 0.30 | 56.58% ± 0.92% | 14754 ± 269 | 8.74 ± 0.23 |
>
> We appreciate the opportunity to strengthen the statistical grounding of our results :)

---

### Note · Authors · 2025-08-12

We sincerely thank the AC and reviewers for their time, feedback, and constructive suggestions.

The **initial reviews (scores: 4, 4, 4, 3)** provided positive feedback on our work, CoPA & HAR, highlighting its novelty (`zBnd, ZCQb`), significance (`zBnd, 6TF1, pJF9, ZCQb`), and quality (`zBnd, 6TF1, pJF9, ZCQb`). Reviewers characterized the paper as well-written (`6TF1, pJF9, ZCQb`), well-structured (`zBnd, 6TF1, ZCQb`), and demonstrating a "high degree of rigor" (`ZCQb`). They praised our method as "comprehensive and ambitious" (`ZCQb`), "technically robust" (`zBnd`), and "conceptually elegant" (`6TF1`), noting its potential to "benefit the broader research community" (`pJF9`). Our empirical results were acknowledged as "rigorous" (`zBnd`), "compelling" (`6TF1`), showing "dramatic improvement" (`ZCQb`), with "strong gains over baselines while maintaining computational efficiency" (`pJF9`).

---

During our discussion, we focused on the following aspects, which will be **updated ([U] in our rebuttal) in our revision**:

- **Enhanced Discussions**. We analyzed the negative impact of dataset denoising on specific benchmarks (`zBnd, 6TF1`), discussed reasons and solutions for HAR's failure modes (`zBnd`), and compared the computational efficiency of CoPA and HAR with baselines (`6TF1, pJF9, ZCQb`).

- **Expanded Evaluation**. We provided 3-run averages and standard deviations for the main results (`zBnd, ZCQb`), conducted test-time scaling experiments of H-BFS and HAR (`pJF9`), included budget-aligned comparisons (`6TF1, pJF9, ZCQb`), and added more evaluations across various benchmarks and base models (`ZCQb`). Our experiments consistently demonstrate the effectiveness and generalizability of our proposed methods.

Additionally, we will incorporate suggestions including more explanations for D-FPS (`zBnd`), more discussions about potential negative social impacts (`zBnd`), future works about generalizability (`6TF1`), further comparisons to related works (`zBnd, pJF9`), discussions on compute requirements (`6TF1`), and writing details (`zBnd, ZCQb`).

---

We appreciate that all reviewers acknowledged our responses, and Reviewer ZCQb will raise the rating. We hope this "comprehensive and ambitious attempt" can "benefit the broader research community." With combined efforts, we can further advance the development of verifiable reasoning in LLMs.

Sincerely, thank you once again to the AC and reviewers! Your feedback has greatly strengthened our paper :)

---

### Decision · Program_Chairs · 2025-09-17

**Decision:**

Accept (poster)

**Comment:**

I recommend accepting this paper, which introduces two novel contributions to address challenges in Deductive Formal Problem-Solving (D-FPS). The Hierarchical Autoregressive Formal Reasoner (HAR) decouples informal-aligned drafting from detailed proving, formulating solution construction as autoregressive generation with per-step feedback. Chain-of-Proxy-Autoformalization (CoPA) addresses data scarcity through a cascaded pipeline of statement autoformalization, proof drafting, and proof search. Experiments demonstrate significant improvements: HAR achieves superior performance on FormalMath500 (43.39%) and MiniF2F-Solving (55.8%) while maintaining computational efficiency, and CoPA shows consistent scalability. The authors have addressed reviewer concerns by providing statistical validation through multiple runs, expanding evaluations to additional benchmarks and models, clarifying terminology, and demonstrating their method's generalizability.